# How many grains are needed for quantifying catchment erosion from tracer thermochronology?

Andrea Madella[1], Christoph Glotzbach[1], Todd A. Ehlers[1]

[1]Department of Geosciences, University of Tuebingen, Schnarrenbergstr. 94-96, 72076 Tuebingen, Germany

*Correspondence to*: Andrea Madella (andrea.madella@uni-tuebingen.de)

**Abstract.** Detrital tracer thermochronology utilizes the relationship between bedrock thermochronometric age-elevation profiles and a distribution of detrital grain-ages collected from river, glacial, or other sediment to study spatial variations in the distribution of catchment erosion. If bedrock ages increase linearly with elevation, spatially uniform erosion is expected to yield a detrital age distribution that mimics the shape of a catchment's hypsometric curve. Alternatively, a mismatch between

detrital and hypsometric distributions may indicate spatial variability of sediment production within the source area. For studies seeking to identify the pattern of sediment production, detrital samples rarely exceed 100 grains due to the time and costs related to individual measurements. With sample sizes of this order, detecting the dissimilarity between two detrital age distributions produced by different catchment erosion scenarios can be difficult at a high statistical confidence level. However, there are no established software tools to quantify the uncertainty inherent to detrital tracer thermochronology as a function of

sample size and spatial pattern of sediment production. As a result, practitioners are often left wondering 'how many grains is enough to detect a certain signal?'. Here, we investigate how sample size affects the uncertainty of detrital age distributions and how such uncertainty affects the ability to infer a pattern of sediment production of the upstream area. We do this using the Kolmogorov-Smirnov statistic as a metric of dissimilarity among distributions. From this, we perform statistical hypothesis testing by means of Monte Carlo sampling. These techniques are implemented in a new tool (*ESD_thermotrace*) to: (i)

consistently report the confidence level allowed by the sample size, as a function of application-specific variables, and given a set of user-defined hypothetical erosion scenarios; (ii) analyse the statistical power to discern each scenario from the uniform erosion hypothesis; and (iii) identify the erosion scenario that is least dissimilar to the observed detrital sample (if available). *ESD_thermotrace* is made available as a new open-source Python-based script along with test data. Testing between different hypothesized erosion scenarios with this tool provides thermochronologists with the minimum sample size (i.e. number of

bedrock and detrital grain-ages) required to answer their specific scientific question, at their desired level of statistical confidence.

## 1 Introduction

Tracer thermochronology uses the distribution of single-grain thermochronometric ages from detritus to infer the spatial pattern of erosion in the source area (e.g. Stock et al., 2006; Vermeesch, 2007). This approach is typically applied where bedrock

thermochronometric age data exhibit a clear age-elevation relationship, allowing inference of the relative contribution of source elevations from the detrital grain-age distribution. A detrital grain-age distribution that closely follows the catchment's hypsometric curve (i.e., the cumulative distribution function of elevation area), is generally interpreted as indicative for spatially uniform erosion. Conversely, a detrital age distribution skewed towards younger (or older) ages may be the consequence of focused erosion at lower (or higher) elevations (Brewer et al., 2003). Tracer thermochronology has been shown to be a powerful tool to investigate the sub-catchment-scale variability of denudation. Geomorphologists have been able to infer changes in climatic parameters (Nibourel et al., 2015; Riebe et al., 2015), glacial erosional processes (Ehlers et al., 2015; Enkelmann and Ehlers, 2015; Clinger et al., 2020), sediment dynamics (Lang et al., 2018), relief evolution (McPhillips and Brandon, 2010; Whipp et al., 2009), occurrence of mass-wasting (Vermeesch, 2007; Whipp and Ehlers, 2019) and differences in rock uplift (McPhillips and Brandon, 2010; Glotzbach et al., 2013, 2018; Brewer et al., 2003; Ruhl and Hodges, 2005). Other work has noted that neglecting mineral fertility variations with catchment lithologies may challenge the conclusions of some of these studies (Malusà et al., 2016). Unfortunately, the number of measured detrital ages for tracer thermochronology is often dictated by inherent limitations of the sampled material and/or by available finances, rather than a science-based choice. Detrital sample sizes often range between 40-120 grains (e.g. Stock et al., 2006; Vermeesch, 2007; McPhillips and Brandon, 2010; Ehlers et al., 2015; Riebe et al., 2015; Glotzbach et al., 2018; Lang et al., 2018; Clinger et al., 2020), and are considered to yield high-confidence results when surpassing ~100 grains based on previous work on sediment provenance analysis (Vermeesch, 2004). However, in unfortunate cases, two measured distributions generated from different erosional patterns cannot be statistically discerned at a high confidence level even with more than 100 grains. Although this issue is well-known to the community (e.g. Avdeev et al., 2011) since the early days of such detrital studies (Brewer et al., 2003), there is no established tool to quantify the uncertainty inherent to detrital tracer thermochronology as a function of sample size and upstream pattern of sediment production. Moreover, the number of measured grains may often be based on convenience and/or habit.

Here, we complement previous work by investigating how sample size affects the uncertainty of detrital cooling age distributions and the related confidence in addressing the pattern of sediment production in the upstream area. We discuss the approaches used in previous case studies, upon which we develop a tool (Earth System Dynamics - *ESD_thermotrace*) to consistently report confidence levels as a function of sample size and case-specific variables. We illustrate our approach using a published dataset from the Sierra Nevada Mountains, California (Stock et al., 2006). The proposed tool is made available as a new open-source Python-based script along with test data. We demonstrate how *ESD_thermotrace* can assist future tracer thermochronology studies in defining the necessary sample size to answer their specific scientific question. In cases where larger sample sizes are impossible to achieve, the statistical power of a tracer thermochronology analysis can be studied using our script.

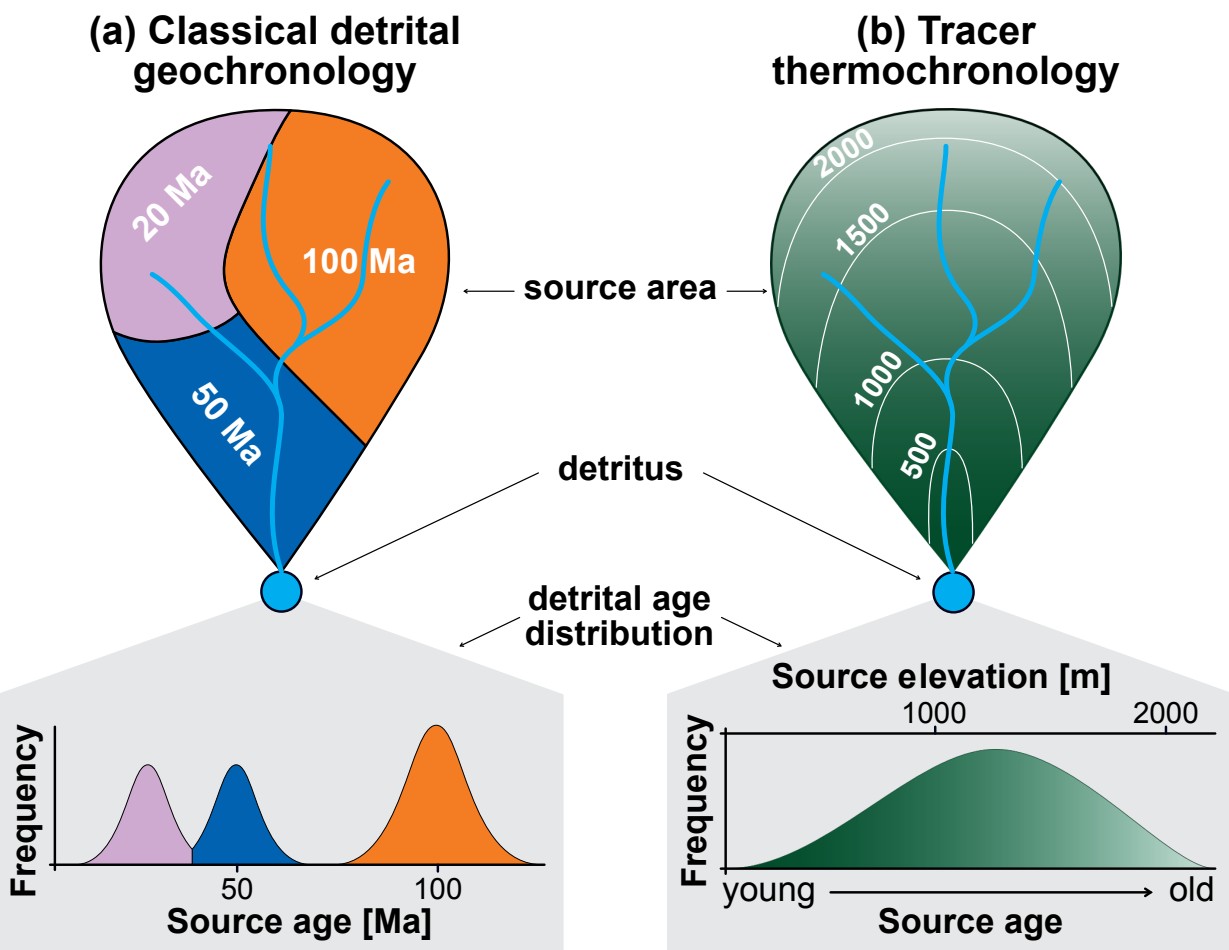

**Figure 1: Sketch of the difference between classical detrital geochronology (a) and tracer thermochronology (b). (a) Discrete age components are found in the detritus and refer to different upstream geological units. (b) A continuous detrital age distribution informs the relative abundance of material sourced from different elevations, based on a known age-elevation relationship.**

## 2 Background information

Single grain detrital age distributions are extensively applied in classical detrital geochronology studies (Hurford and Carter, 1991), where U/Pb crystallization ages of zircon constitute by far the most used tool (Spiegel et al., 2004; Andersen, 2005;

Malusà et al., 2013). In this type of application, the aim is to obtain the spectrum of all age components (i.e. age peaks) that characterize a siliciclastic sediment. If a range of assumptions hold (Malusà et al., 2013; Malusà and Fitzgerald, 2020), the provenance of a sediment sample's source area can be inferred by matching the detrital age components to those of known upstream geological units and/or events. For that purpose, the number of measured detrital grains determines the confidence of detecting minor/small age components. An exhaustive probabilistic method to report such confidence exists (Vermeesch,

2004) and is not the focus of this study. The absolute age components of the source area are in fact unimportant in detrital

tracer thermochronology (Avdeev et al., 2011), for which monolithologic catchments are best suited, in order to minimize mineral fertility issues in the source rock (Fig. 1). The focus here is the dissimilarity between the distribution of ages found in the source area and in the fluvial/glacial/hillslope sediment derived therefrom, regardless of their absolute age components. For this purpose, the uncertainty caused by a small sample size strongly limits the least significant dissimilarity that can be resolved between two distributions. This minimum resolution directly affects the power of our inferences.

In the following we summarize the conceptual model concerning this matter and the approaches that have been used to address it thus far. Let us consider a monolithologic catchment and a set of detrital grain-ages measured at its outlet. The observed grain-age distribution should match a predicted distribution that is forward-modelled stacking the following layers of geologic information about the upstream area (Fig. 2):

I. The catchment hypsometry, or the distribution of the catchment's cumulative area as a function of elevation, which is derived from the digital elevation model (DEM) of the study area, and it has a negligible uncertainty.

II. The bedrock age-elevation data. In the simplest case, a linear function based on a dataset of ages with associated uncertainty, measured at known elevations.

III. The mineral fertility, which informs the original abundance of the target mineral in the different elevation ranges and is mostly a function of lithologic variability. This is a critical parameter that can lead to gross misinterpretations if ignored (Malusà et al., 2016).

IV. Information on how the sediment-size distribution varies between the headwaters and the detritus. This is to make sure that grains in the detritus are representative of erosion in the catchment (e.g. Vermeesch, 2007; Riebe et al., 2015; Lukens et al., 2020).

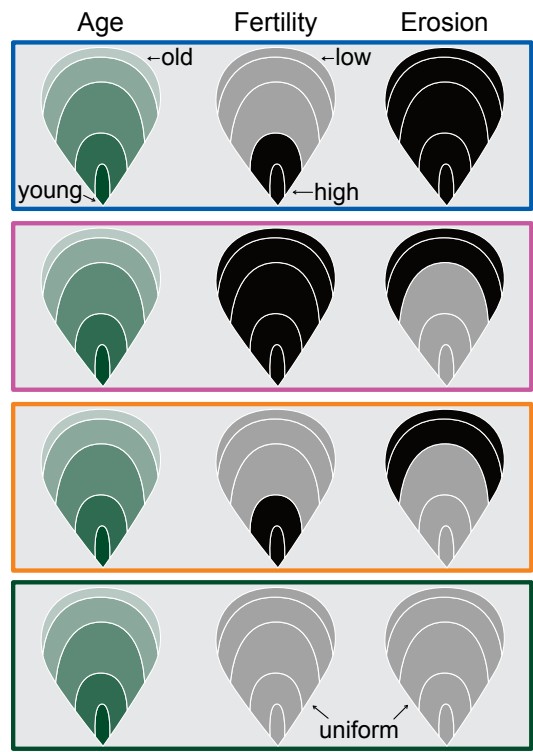

**(a) Catchment data**

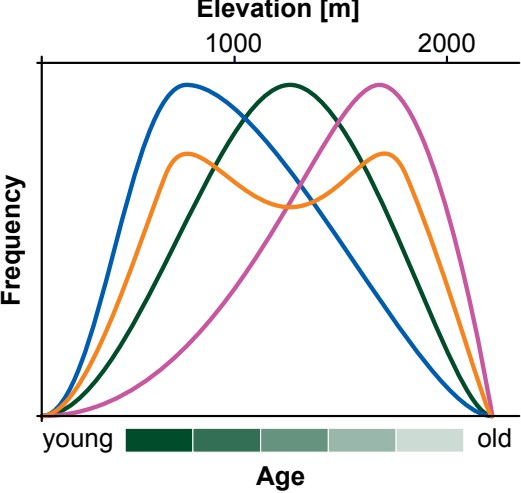

**(b) Detrital data**

Figure 2: Qualitative sketch to illustrate the effect of mineral fertility and erosion on the detrital distribution. (a) The catchment of Fig. 1b with known bedrock age (shades of green) is subject to 3 scenarios of spatially varying fertility and erosion. The box outlines refer to the curves below. (b) Detrital distributions obtained from the different scenarios in above. The green curve refers to spatially uniform erosion and fertility.

V.     A pattern of erosion across the catchment, which is the dependent variable of the system. This is any map of erosion probability, a scenario to be tested against uniform erosion (or other reference scenarios).

Depending on the scope of the application, tracer thermochronology ultimately aims to quantify the mismatch between observed and predicted distributions, where the predicted distributions vary depending on assumed models for catchment erosion, sediment dynamics and tectonics.

## 3 Comparing predicted and observed distributions

Predicted distributions should be constructed accounting for all above information and related uncertainties, such that the confidence of the fit to the observed distribution also accounts for them. Brewer et al. (2003), and later Ruhl and Hodges (2005), were the first to compare the distribution of detrital thermochronometers to that of age-elevation data. Although the scope of their work differed from more recent tracer thermochronology, the evaluation of the dissimilarity between predicted/observed distributions remains the main object of their studies. These authors constructed synoptic probability density functions (SPDF) of the observed data by "stacking" the gaussian distributions of all measured grain-ages, each with their analytical error (this is equivalent to the $SPDF_t$ in Table 1 of Vermeesch (2007)). In addition to the observed distribution, a predicted SPDF was also constructed with the same method, where the predicted grain-ages are a random subsample of the hypsometric curve and are each given an arbitrary average uncertainty. Brewer et al. (2003) define the mismatch $P_{diff}$ between observed and predicted SPDF with equation (1):

$$P_{diff} = \frac{\sum_{t=0}^{t=\infty}|P_1(t)-P_2(t)|}{2} \times 100, \tag{1}$$

where ($P_1$ and $P_2$) are the probabilities of the two distributions calculated at each age step ($t$). $P_{diff}$ relates to the area comprised between two SPDF in the age-frequency space. Brewer et al. (2003) calculate the 95% confidence mismatch between observed and predicted SPDF through a Monte Carlo simulation.

Vermeesch (2007) has shown that, for the purpose of comparison, observed and predicted distributions are best expressed as a cumulative age distribution (CAD) rather than SPDF. A CAD is a step-function with the sorted mean ages on the x axis and the related quantiles on the y axis (Vermeesch, 2007). This method is preferred because it avoids the possible sources of bias introduced by: (i) choice of the smoothing parameter in the kernel density estimations (KDE); (ii) binning in histograms; and (iii) uncertainty-based weighting in the SPDF curves (Vermeesch, 2012). To evaluate the goodness of fit between observed and predicted CADs, Vermeesch (2007) uses the Kolmogorov-Smirnov (KS) statistic, which informs the maximum distance $d_{KS}$ between two cumulative distribution functions as follows (Massey, 1951 and references therein):

$$d_{KS} = maximum\left|CAD_{observed}(t) - CAD_{predicted}(t)\right|. \tag{2}$$

Given an observed CAD with $k$ observations (i.e., ages), $d_{KS}$ is calculated for several $n=k$ sub-samples of the predicted CAD. The 95th percentile of all sorted $d_{KS}$ ($d_{KS\_95}$) is used as the least significant dissimilarity to reject the null hypothesis that the observations are drawn from the predicted CAD. In other words, an observed CAD that plots entirely within the range

$CAD_{predicted} \pm d_{KS\_95}$ (Fig. 3) cannot be discerned from the predicted age population at the 95% confidence level. As an alternative to this iterative method, the confidence region for a predicted CAD can be calculated with the analytical solution of the Dvoretzky-Kiefer-Wolfowitz (DKW) inequality as follows:

$$d_{KS\_95} \approx \varepsilon = \sqrt{\frac{\ln\frac{2}{\alpha}}{2k}}, \tag{3}$$

where the DKW distance $\varepsilon$ approximates well the $d_{KS\_95}$ as a function of the confidence level $(1 - \alpha)$ and the sample size $k$ (Fig. 3) (Massart, 1990; Massey, 1951).

Riebe et al. (2015) further developed the bootstrapping approach described above to age distributions (in SPDF form). Instead of basing their analysis on the KS statistic, the 95% confidence envelope of the prediction is iteratively estimated at each age step $t_i$. For each $t_i$, the distribution of 10,000 predicted age frequencies $SPDF_{predicted}(t_i)$ is used to draw 2.5th and 97.5th

percentiles at each age step. Because the age steps $t_i$ relate to the elevation steps in the catchment through the bedrock thermochronometric data, those elevation steps exhibiting excess (>97.5th) or deficit (<2.5th) frequency are interpreted to produce sediment in excess or deficit with respect to the reference scenario of uniform erosion, at the 95% confidence level.

The approaches summarized above are well-suited to test observed detrital age distributions against the null hypothesis of

spatially uniform erosion. However, even if the uniform erosion hypothesis is rejected, such a test does not yield further

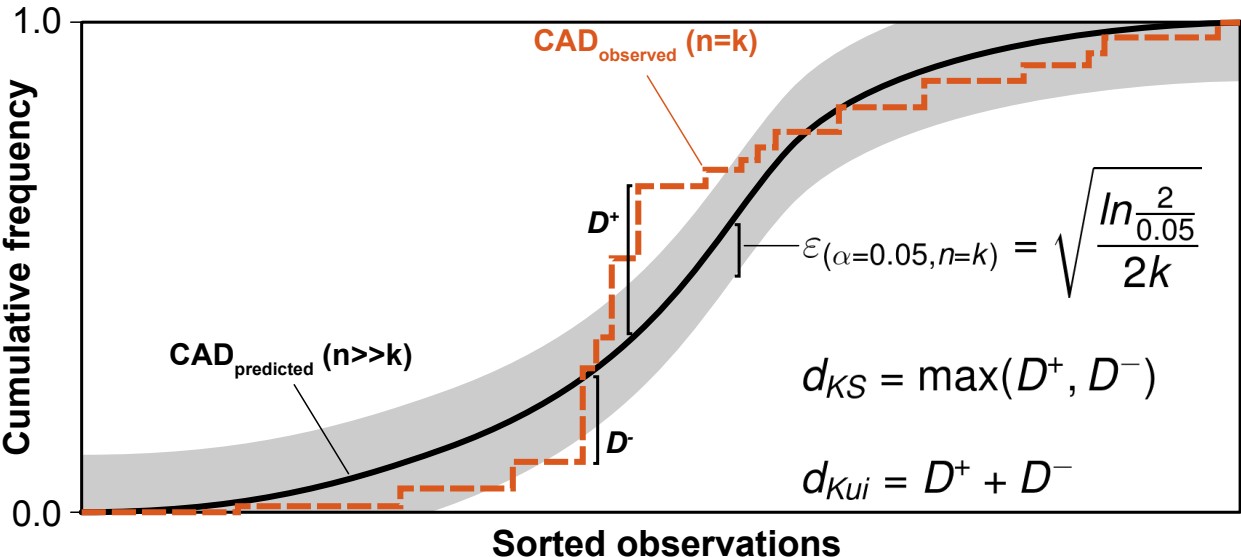

**Figure 3: Example of metrics to compare two cumulative distributions drawn from different sample sizes.** Here, an observed Cumulative Age Distribution (CAD) for n=k (orange stepwise dashed line) is compared to a predicted CAD drawn for n>>k (solid black line). The Kolmogorov-Smirnov statistic $d_{KS}$ equals the maximum absolute vertical distance between distributions in the observations-frequency space. The Kuiper statistic $d_{Kui}$ equals the sum of both positive and negative maxima. For $\alpha$=0.05 (i.e. 95% confidence) and n=k, a Dvoretzky-Kiefer-Wolfowitz confidence region can be calculated (gray shading). In this example, observed and predicted distributions are drawn from two statistically different populations at the 95% confidence level, because the orange curve exceeds the gray region.

information about the spatial variability of sediment production. One way to gain additional information would be to test the observed distribution against predicted age distributions from erosion scenarios other than spatially uniform, thereby quantifying the likelihood that the measured grain-ages could be produced by the tested scenarios. This approach would help practitioners in deciding the sampling strategy, calculating the appropriate number of measurements and the resources to be allocated for them. In the next paragraphs we introduce the new *ESD_thermotrace* software, an open-source tool built on the cited previous work to consistently perform the following: (i) determine the confidence level allowed by the detrital sample size in rejecting the uniform erosion hypothesis; (ii) analyze the statistical power of detecting the effect size caused by alternative erosion hypotheses, as a function of the number of grains; and (iii) test observed detrital distributions against all input erosion hypotheses (both uniform and alternative scenarios) and calculate the likelihood that the detrital sample may have been drawn from either of them. The software is introduced in the paragraphs below.

## 4 ESD_thermotrace: A tool to study the uncertainty of tracer thermochronology datasets

The software *ESD_thermotrace* (Madella et al., 2022) performs the steps briefly outlined here below. For additional details, the reader is referred to the illustrative case study farther below and the well-commented code itself (https://doi.org/10.5880/fidgeo.2021.003):

1. Bedrock age map interpolation.
   - Input: bedrock age-elevation dataset and digital elevation model.
   - Output: Bedrock age map.
   - Method: users can choose among 1D linear regression, 3D linear interpolation, 3D radial basis function. Alternatively, an externally produced age map can be imported.
2. Bedrock age uncertainty map interpolation.
   - Input: bedrock age-elevation dataset with uncertainties (point data) and bedrock age map (grid data).
   - Output: Bedrock age uncertainty map.
   - Method: The uncertainty of the age map is estimated through bootstrapping. An externally produced uncertainty map is required if the age map was imported.
3. Extract catchment bedrock age, coordinates, mineral fertility and erosion data.
   - Input: catchment outline, bedrock age and age uncertainty maps, mineral fertility map, one or more erosion maps.
   - Output: a table of all the listed catchment properties necessary to predict detrital age distributions
   - Method: for each cell of the age map contained in the catchment outline, the local coordinates, age, fertility and erosional weight(s) are extracted.
4. Predict detrital grain age distributions for each erosion scenario.
   - Input: table of catchment data.
   - Output: a predicted detrital age population for each erosion scenario and related cumulative age distribution.

o   Method: An amount of ages proportional to erosional weight and fertility is drawn from each cell, for each scenario. The ages from all catchment cells collectively represent a predicted population, from which cumulative age distributions are constructed.

5. Calculate (i) the likelihood of rejecting the uniform erosion hypothesis with the observed sample size, and (ii) the statistical power of discerning predicted erosion scenarios (i.e. alternative hypotheses) as a function of sample size.

o   Input: one or more sets of observed grain-ages and uncertainties, the predicted detrital populations and distributions.

o   Output: A graph displaying: (i) the confidence of rejecting uniform erosion with the observed sample size; and (ii) the statistical power curve of discerning the scenarios from uniform erosion varying sample size.

o   Method: first, the $d_{ks\_95}$ for the available sample size $k$ is calculated with Equation 3; then, the likelihood that the $n=k$ observed CAD is more dissimilar than the $d_{ks\_95}$ is calculated through bootstrapping. The same operation is also repeated for a range of sample sizes ($20<k<130$), in order to estimate the rise in confidence level caused by the increasing sample size (if the observed distribution and associated uncertainty remained identical despite the changing sample size). To calculate the statistical power of discerning the tested erosion hypotheses, the same approach is applied. In this case, however, the software draws a number of distributions from each erosion scenario, instead of the observed grain-ages.

6. Given a set of observed detrital grain-ages and uncertainties, calculate the plausibility of each erosion scenario (i.e. the likelihood that alternative hypotheses cannot be rejected with high confidence).

o   Input: same as point 5).

o   Output: two plots to visualize how plausibly the observed grain-ages have been drawn from the predictions.

o   Method: in the first plot, dissimilarities calculated between predictions and observations are sorted and their distribution is plotted in the form of a violin plot. Second, a two-dimensional Multi-Dimensional Scaling (MDS) model is fitted to the dissimilarities among the predicted and observed distributions and plotted following Vermeesch (2013).

All the above operations are embedded in an open-source Jupyter Notebook (jupyter.org), a software that allows integrating text, Python code, and visualizations within the same document for maximum editability and transparency. All plots are produced with Matplotlib (Hunter, 2007) and Seaborn (Waskom et al., 2020) Python libraries and are colored using the color blind-friendly and perceptually uniform ScientificColourMaps6 (Crameri et al., 2020). In the following paragraph, for illustrative purpose only, we show how the program helps analyzing an already published detrital AHe age dataset (Stock et al., 2006).

**5 Application of ESD_thermotrace to the Inyo Creek case study**

We apply *ESD_thermotrace* to the bedrock and detrital apatite (U-Th(-Sm)/He (AHe thereafter) age datasets of Stock et al. (2006). In that study, 9 bedrock AHe ages from the Inyo Creek and adjacent Lone Pine Creek catchments (eastern Sierra

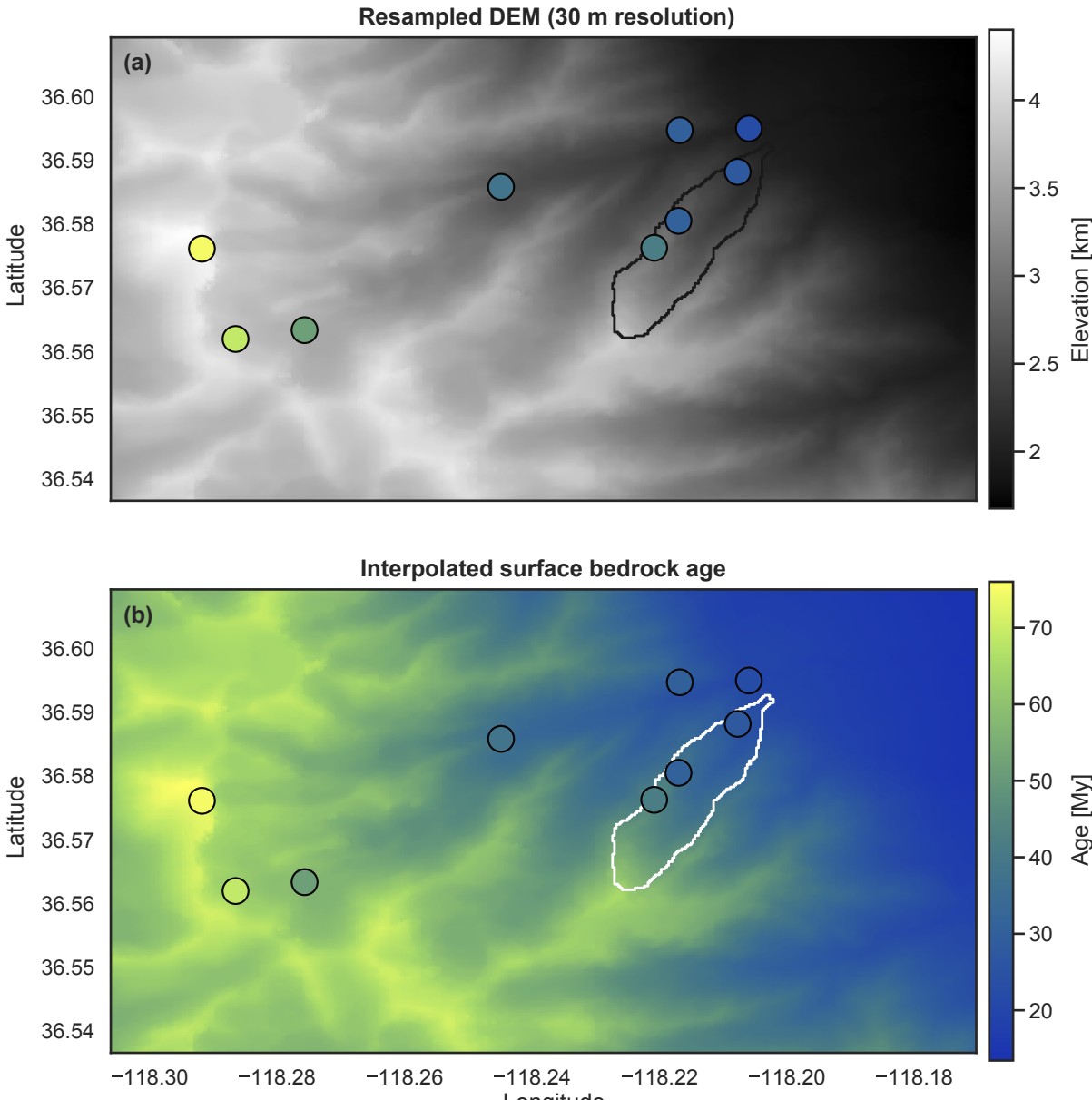

**Figure 4: Input bedrock data from Stock et al. (2006). (a) Raster images of the study area's digital elevation model, resampled to the user-specific cell size, and (b) bedrock surface AHe age interpolated based on the linear regression of age-elevation data. In both plots, point data inform bedrock sample locations and related AHe ages. Polygons show the location of the Inyo Creek catchment.**

Nevada, California, USA) are used to constrain the age-elevation relationship of the source area. The authors compared these bedrock data to the AHe age distribution of river sand samples from both catchment outlets. For the sake of simplicity, here we only consider the fine sand sample ($k$=52) from the Inyo Creek catchment. Based on their analysis Stock et al. (2006) 220 inferred no significant variability of sediment production (erosion) across all elevations within the catchment.

**5.1 Data import and bedrock age interpolation**

The first few steps of *ESD_thermotrace* allow importing: (i) the digital elevation model of the study area; (ii) the polygon of the catchment of interest; (iii) a table of bedrock age, age uncertainties, and elevation data; (iv) the (optional) erosion scenarios to be tested; and (v) a (optional) fertility map. Next, the surface bedrock ages are calculated with the preferred method (see above and README file in Madella et al., 2022). The bedrock AHe ages of Stock et al. (2016), recalculated after Riebe et al. (2015), exhibit a ~60 My age increase from ca. 2 to 4.5 km of elevation range that is well-described by an inverse variance-weighted linear regression ($r^2 = 0.93$). The input data are plotted in Figure 4a. In Figure 4b, every cell is assigned an AHe age as a linear function of elevation (Fig. 5b), to map the bedrock cooling age on the topographic surface. The interpolation error is also mapped in Figure 5a, and it displays the $1\sigma$ uncertainty of the prediction from the linear regression (Fig. 5b).

**5.2 Extraction of catchment data**

Next, *ESD_thermotrace* extracts the x,y,z coordinates, the bedrock cooling age and the related error for all the cells bound by the catchment outline. These data are written in a table, to which a column informing erosional weights is added for each desired erosion scenario and for mineral fertility. In addition to the user-defined erosion maps, by default the software considers the uniform erosion scenario *'Euni'* (spatially constant erosional weight). Two further example scenarios can be toggled to test for an exponential increase of erosion with elevation or an exponential decrease of erosion with elevation (not considered here). We note that any other spatial variation in erosion can be defined by a user such that 'erosion maps' for the catchment

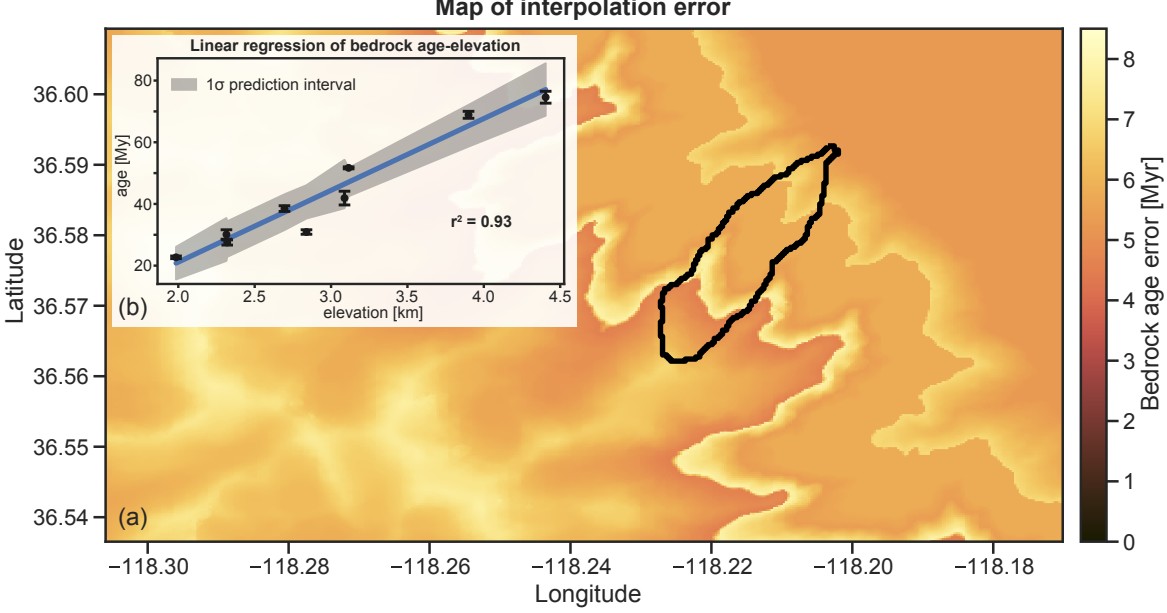

**Figure 5:** The raster image (a) displays the bedrock age interpolation error mapped to the topographic surface of the study area. The inset (b) shows the inverse variance-weighed linear regression employed as age-elevation function, as well as the associated prediction interval with $1\sigma$ confidence level.

following stream power, slope dependent, glacial slide velocity, or other approaches from geomorphic transport laws can be input.

For this case study, bulk geochemistry and point-counting analyses of Hirt (2007) indicate that apatite fertility does not

significantly vary within the three lithologies found in the Inyo Creek catchment (Lone Pine granodiorite, Paradise granodiorite, Whitney granodiorites). For illustrative purpose, we test uniform erosion against two opposite step functions of elevation: a scenario with F-times higher erosion efficiency above the median catchment elevation and one with F-times higher

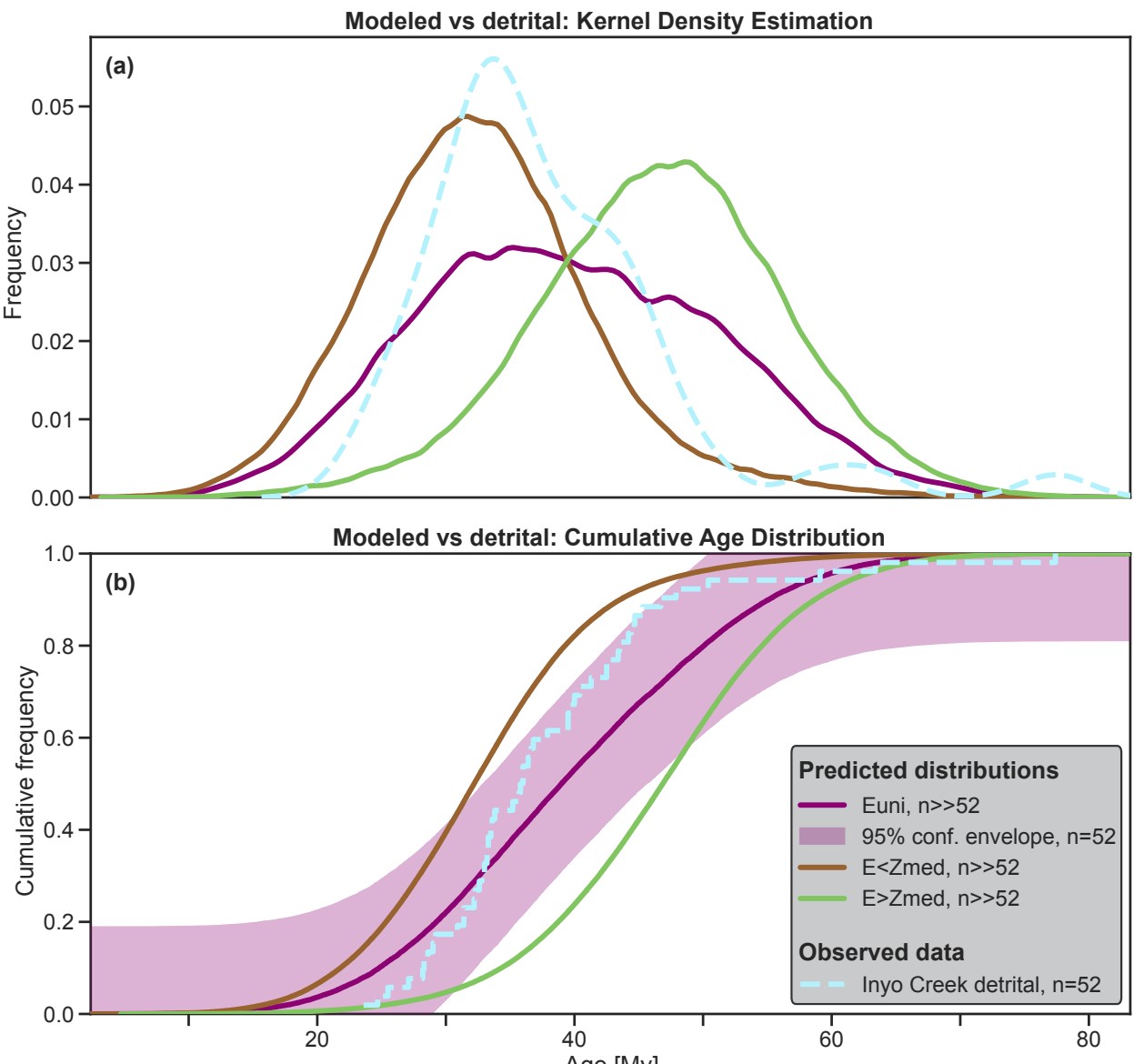

**Figure 6: Kernel Density Estimation curves of all predicted detrital distributions as well as of the observed detrital sample (a). Cumulative form (b) of the same distributions, plus the 95% confidence envelope (DKW-bounds) of 'Euni' for n=k=52. Euni: uniform erosion; E<Zmed: tenfold erosion below median elevation; E>Zmed: tenfold erosion above median elevation.**

erosion below the median elevation. The factor F equals twice the ratio between the most frequent and the least frequent elevation. For this calculation, the hypsometric histogram is constructed with a number of bins equal to the maximum difference in bedrock cooling ages, divided by the mean age uncertainty (rounded up to the next integer). Here, the Inyo Creek catchment is binned in 7 elevation ranges, of which the most frequent is 5-times the least frequent one, resulting in F = 10. In other words, we test for an increase in erosion twice as prominent as the hypsometric peak (10-fold), once below ('*E<Zmed*') and once above ('*E>Zmed*') the median catchment elevation.

### 5.3 Prediction of age populations and detrital age distributions

The erosional weights are used to forward model sediment production at each position in the catchment. To do so, an amount of grain-ages proportional to the erosional weight and to the local mineral fertility are randomly chosen for each cell. These grain-ages are randomly drawn from a normal distribution constructed using the local interpolated bedrock age and the related uncertainties. The randomly picked grain-ages of all cells are stored in one suite of grain-ages, which represents the predicted detrital grain-age population of a well-mixed fluvial sediment at the catchment outlet. Such a detrital population is predicted for each erosion scenario, and related predicted CADs are constructed by sorting the age populations and plotted as a step function (Fig. 6b). For the sake of quick visual comparisons, *ESD_thermotrace* plots a kernel density estimation with arbitrary smoothing (Fig. 6a). However, the dissimilarity among distributions will be evaluated exclusively based on the cumulative form, in order to minimize bias (Vermeesch, 2007). In Figure 6b, all predicted and observed cumulative age distributions are plotted. For a first order impression of the uncertainty due to sample size, the 95% confidence envelope of the reference scenario is also calculated with Eq.3 and plotted in the background. Here, the user can specify for which reference scenario ("Euni" as default) the confidence envelope should be plotted.

### 5.4 Confidence level and statistical power as a function of sample size

After predicting the distributions, to answer the question "how many grain-ages are necessary to discern a hypothetical erosion pattern from uniform erosion?" the software analyzes the statistical confidence and the statistical power as a function of sample size. The confidence informs the likelihood of rejecting a null hypothesis $H_0$ (the uniform erosion scenario) based on the observations (the dated grains). The statistical power informs the likelihood that, if an alternative hypothesis $H_1$ was true (one of the tested erosion scenarios), $H_0$ would be rejected based on a sample drawn from $H_1$. Here, the software calculates: (i) the maximum confidence in rejecting the uniform erosion hypothesis allowed by the observed sample size; and (ii) the statistical power of discerning the scenarios from uniform erosion as a function of sample size. Let us consider the observed CAD constructed with the sorted mean grain-ages. The mean standard deviation of the observed grain-ages is used to generate 10'000 Monte Carlo samples of this observed CAD, which account for age dispersion due to analytical error and/or reproducibility. Then the confidence in rejecting uniform erosion equals the fraction of sampled CADs that is more dissimilar to 'Euni' than the least significant dissimilarity ($d_{KS\_95}$) allowed by the sample size $k$ (see Eq.3). The so-calculated statistical confidence can be read in the scatterplot of Figure 7 (light blue circle), and it equals 55% for the Stock et al. (2006) data.

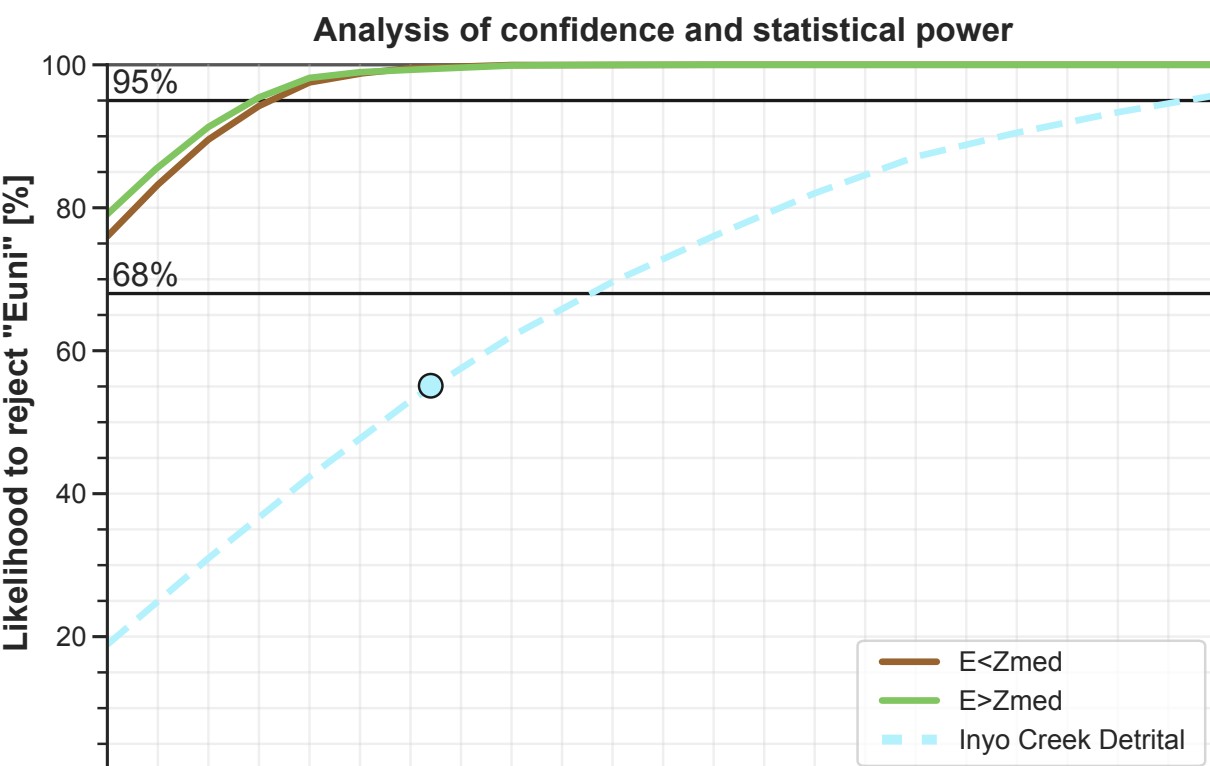

**Figure 7: Confidence level at which the observed CAD (n=52) can be discerned from the uniform erosion prediction (cyan circle). The dashed cyan line shows how the allowed confidence level would vary, if the same observed CAD relied on 20-130 grain-ages. Statistical power of discerning the tested erosion scenarios from 'Euni' as function of sample size (solid lines). In this case study, the observed CAD of the Inyo Creek detrital sample allows rejecting the uniform erosion hypothesis with 55% confidence at best. An analysis to detect the scenarios 'E<Zmed' and 'E>Zmed' would have a statistical power greater than 95% in rejecting 'Euni' with more than 37 grain-ages. Euni: uniform erosion; E<Zmed: tenfold erosion below median elevation; E>Zmed: tenfold erosion above median elevation.**

Figure 7 also shows that the 95% level of significance could be achieved with more than 128 grain-ages. Here, we clarify that the latter estimate assumes that the observed distribution would not change in shape while varying the sample size, and as such it can only be treated as an indicative number.

If the confidence allowed by the actual sample size is lower than the previously chosen level of significance, the analysis of

statistical power can help identifying how likely the available number of grain-ages could detect a known effect size (i.e. the dissimilarity caused by a known erosion scenario). This analysis of statistical power is based on the iterative comparison between the reference scenario and all other erosion scenario predictions. This is achieved through random subsampling of the predicted distributions for a range of possible sample sizes ($20 \leq n \leq 130$ in this case). For each $n_i$, the least significant dissimilarity $d_{KS\_95}(n_i)$* of the reference scenario is calculated for $\alpha = 0.05$ and $n = n_i$ using Equation 3. In the observations-

cumulative frequency space (Fig. 6b), $d_{KS\_95}(n_i)^*$ represents the vertical distance that any $n=n_i$ distribution needs to exceed to be discerned from a $n=n_i$ subsample of the reference scenario, with 95% confidence (purple shading in Fig. 6b). Next, 1,000 $d_{KS\_95}(n_i)$ are calculated between $n=n_i$ subsamples of each erosion scenario and the reference scenario. The probability of $d_{KS\_95}(n_i)$ exceeding $d_{KS\_95}(n_i)^*$ for each $n_i$ can be read from the curves in Figure 7. For the Inyo Creek case study (Stock et al., 2006) both tested erosion scenarios ('$E<Zmed$', '$E>Zmed$') yield predicted CADs that are very dissimilar from uniform erosion. Consequently, the statistical power to discern these alternative erosion hypotheses would exceed 95% even with only 40 grains.

Figure 7 shows the use of *ESD_thermotrace* as a tool to explore the feasibility of a tracer thermochronology study. Based on their research question, users can apply a few possible erosion maps and test the likelihood with which the respective detrital distributions could be discerned from uniform erosion through detrital tracer thermochronology, as a function of sample size. Such analysis of feasibility is not only beneficial in terms of better quantifying uncertainties, but it can also assist investigators in defining the budget for measurements at an early stage of proposal writing. Alternatively, in cases where the number of datable grains is limited by material properties, budget, or other logistic reasons, Figure 7 informs the maximum confidence level of the results.

## 5.5 Evaluating the plausibility of test scenarios.

The final steps of *ESD_thermotrace* assist users in finding the erosion scenario that is most likely to generate a predicted CAD that resembles the observed CAD. For each erosion scenario, $d_{KS}$ and $d_{Kui}$ (Fig. 3) are calculated between 10,000 n=k subsamples of the respective predicted CAD and the observed CAD. The distribution of these $d_{KS}$ and $d_{Kui}$ values is shown in the form of a split violin plot in Figure 8a and it is to be compared to the range of values shown by the violin in Figure 8b. The latter shows the distribution of $d_{KS}$ and $d_{Kui}$ calculated between random subsamples of the observed CAD that account for the mean analytical error and the observed CAD itself (constructed only with the mean ages). In other words, Fig. 8b displays how the dispersion of a predicted CAD (accounting for analytical error and sample size) is distributed. The "plausibility" of each scenario is plotted beneath each violin and it equals the probability that the values plotted in Figure 8a fall within the one-sided lower 95th percentile of those shown in Figure 8b. For the sake of clarity, we note that the term "plausibility" used here is equivalent to the false negative rate ($\beta$) in statistical jargon. Accordingly, there is 51.6% probability that the observed CAD could be drawn from the uniform erosion scenario, and 13.7% and 0.0% probability that scenarios '$E<Zmed$' and '$E>Zmed$' generate predicted CADs that fall within the spread of the observed CAD, respectively.

Lastly, as suggested by Vermeesch (2013), the *ESD_thermotrace* program applies a two-component Multi-Dimensional Scaling model (MDS) to all predicted and observed distributions. This algorithm fits a 2-dimensional coordinate system to the measured dissimilarities $d_{KS}$ among all considered distributions (both predicted and observed). In a well-fitted MDS model, distances among points in the modelled 2D-space are a good approximation of the actual dissimilarities among the input

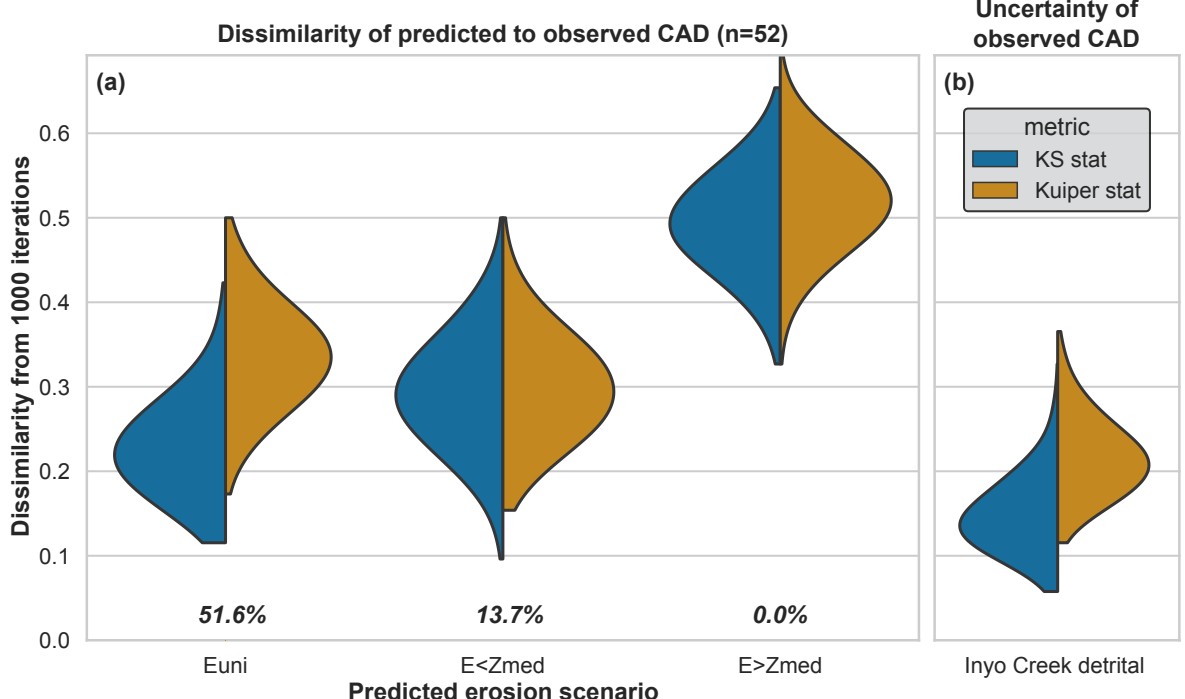

**Figure 8: Violin plot showing the KS (and Kuiper) statistics between the predicted CAD of each erosion scenario and observed CAD, calculated for 1000 n=k=52 subsamples (a). KS (and Kuiper) statistics between the observed CAD and 1000 random n=k=52 distributions, drawn from the same detrital age population but including uncertainty (b). Every violin is split in two halves with equal area, each representing the probability distribution function of 10,000 KS (or Kuiper) statistics. The widest point of each semi-violin informs the median KS (or Kuiper) statistic; The closer to zero, the more similar is a scenario to the observed detrital age distribution. The most unlikely scenario is 'E>Zmed'. Both other scenarios are plausible, because their dissimilarity to the observed CAD largely overlaps the range of dissimilarity due to analytical error of the detrital ages.**

elements (Fig. 9a). To reach a satisfactory fit, modelled dissimilarities are plotted against input dissimilarities (Fig. 9b) and the sum of all distances to the 1:1 line is minimized (a procedure commonly referred to as *stress* minimization). The MDS plot

renders an immediate visualization of the dissimilarity among distributions, where more similar distributions plot closer to each other. Moreover, with the addition of the 68% and 95% confidence ellipses, the degree of overlap among distributions is also easily visually assessed (Fig. 9a).

The application of the *ESD_thermotrace* software to Stock et al.'s (2006) Inyo Creek data shows the following:

-    A sample size of 52 grain-ages allows a maximum confidence level of ~55% in rejecting the uniform erosion hypothesis 'Euni' (cyan circle in Fig. 7). If the observed CAD had been drawn from ~130 grain-ages, this confidence level would exceed 95% (dashed cyan curve in Fig. 7).

-    52 grain-ages would suffice to detect the effect size caused by 'E<Zmed' and 'E>Zmed' (green and brown curves in Fig. 7). However, the observed detrital sample is extremely unlikely to have been drawn from either of these scenarios.

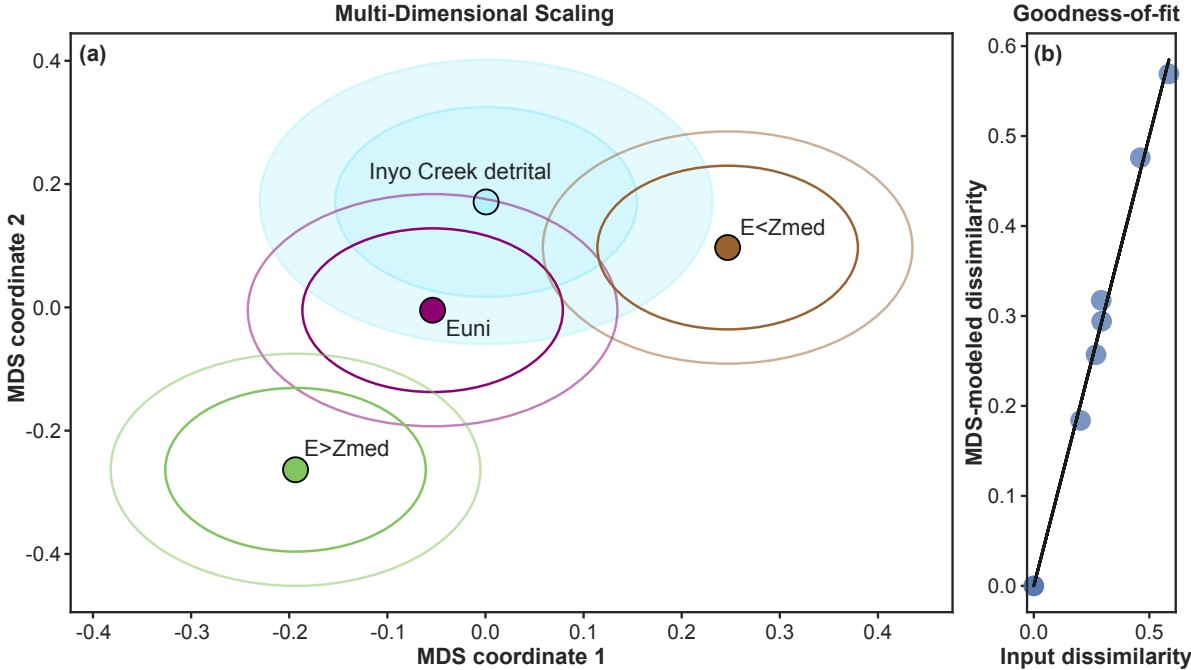

**Figure 9: Multi-Dimensional Scaling (MDS) plot. The 2-components MDS model fits a coordinate system to the measured dissimilarities (i.e. KS statistics) among all distributions. If the MDS model has a good fit, distances calculated in the new 2D-space (a) are a good approximation of the input dissimilarities among the elements. The goodness of fit is shown in (b), where modelled dissimilarities are plotted against input dissimilarities. For each distribution in (A) 68% and 95% confidence ellipses are also plotted.**

-      Among the tested scenarios, the uniform erosion hypothesis 'Euni' is the least dissimilar (51.6% likelihood) (Fig.8 and Fig.9).

Here, while we have shown the functioning of ESD_thermotrace using the Inyo Creek data from Stock et al. (2006), we refrain from proposing a revised interpretation of the catchment's erosion dynamics. We do note that, in addition to the data from Stock et al. (2006), Riebe et al. (2015) have accounted for [10]Be-derived denudation rates and 73 additional AHe ages from

coarse sand-sized sediment. These authors have shown that erosion in the Inyo Creek catchment is best explained by an exponential increase of sediment production with elevation. Thereby, they demonstrate that understanding the pattern of erosion in this catchment requires taking multiple sediment sizes into account (Riebe et al., 2015; Lukens et al., 2020; Sklar et al., 2020; see also discussion below).

**6 How many grains do we need for tracer thermochronology?**

The analysis with *ESD_thermotrace* shows that the appropriate sample size for a tracer thermochronology study cannot be determined a-priori without knowing: (i) the case-specific scientific question (i.e. the spatial pattern of erosion to be tested); (ii) the source area hypsometry; (iii) the desired minimum confidence level; and (iv) the surface bedrock ages and uncertainties. On this note, if possible, it is advisable to explore the feasibility of a study with the already available bedrock ages before

sampling. In absence of available published ages, bedrock samples are best processed first to avoid wasting resources on
potentially inconclusive analyses of detrital grains. To better illustrate the importance of the initial knowledge of the
catchment's geology, we conducted a set of simulations with the same inputs as the Inyo Creek case study (Stock et al., 2006).
In these simulations we vary the location (in terms of elevation range) of maximal erosion and observe the impact on the
statistical power in detecting the imposed pattern. A broad and a narrow gaussian curve of erosion ($1\sigma$ = 500 m and 100 m,
respectively) are applied to the catchment's range of elevations, shifting the position of the peak in 100-m-steps at each
simulation. We test two sets of gaussian functions of elevation (2x and 5x), computed in such a way that the prominence of
peak erosion equals two and five times the prominence of the catchment's hypsometric peak, respectively. Therefore, because
the most frequent elevation range is ca. five times the least frequent bin (Fig. 10c,f), the "2x" and "5x" curves define an erosion
efficiency where peak sediment production is ten times and twentyfive times higher than the minimum in the catchment,
respectively. Results from this parameter study, example erosional functions, and the hypsometric histogram are illustrated in
Figure 10.

Figure 10 shows how the statistical power in detecting these gaussian-peak-scenarios is affected by: (i) the elevation of the
erosional peak; (ii) its amplitude; (iii) its width; and (iv) the detrital sample size. An increase in sample size and/or an increase
in amplitude of the erosional peak always correspond to higher statistical power. The elevation of the erosional peak affects
the statistical power depending on the position of the erosional peak relative to the peak of hypsometry (Fig. 10). In the case
of a broad gaussian function of erosion ($1\sigma$ = 500 m), minima (blue areas) are observed where the erosional peak is located at
elevations straddling 2,800 m (Fig. 10a,b). These minima in statistical power at ~2,800 m coincide with the broad frequency
peak of catchment elevations (Fig. 10c). In this elevation range, the diffused increase in erosion efficiency does accentuate the
peak of the hypsometric curve, but it results in a limited effect size (i.e. a small $d_{KS}$ dissimilarity to "Euni"). With this
configuration, all erosion scenarios peaking within ca. 2,700-2,900 m produce detrital distributions whose effect size is
detected in only <60% of the simulations, regardless of sample size and peak amplitude. This implies that certain combinations
of erosional pattern and distribution of bedrock ages are poorly suited to be investigated by means of tracer thermochronology.

In the case of a narrow gaussian function of erosion ($1\sigma$ = 100 m) (Fig. 10d,e,f), minima of statistical power are additionally
found at peak elevations higher than 3,600 m and lower than 2,200 m (Fig. 10d,e). These occur because peak erosion is applied
to a narrow elevation range that is not frequent enough to produce a statistically relevant number of grains. However, in the
case of narrow erosional peaks, an increase in sample size substantially increases the statistical power to detect erosion
scenarios, even if centered at the critical elevation of ca. 2,700-2,900 m described just above (Fig. 10d,e). This parameter study
demonstrates the importance of analyzing the catchment hypsometry and testing erosion scenarios even before collecting data,
in order to make an informed choice on the appropriate sample sizes and to identify possible scenarios that are unlikely to be
detected with high confidence through detrital tracer thermochronology.

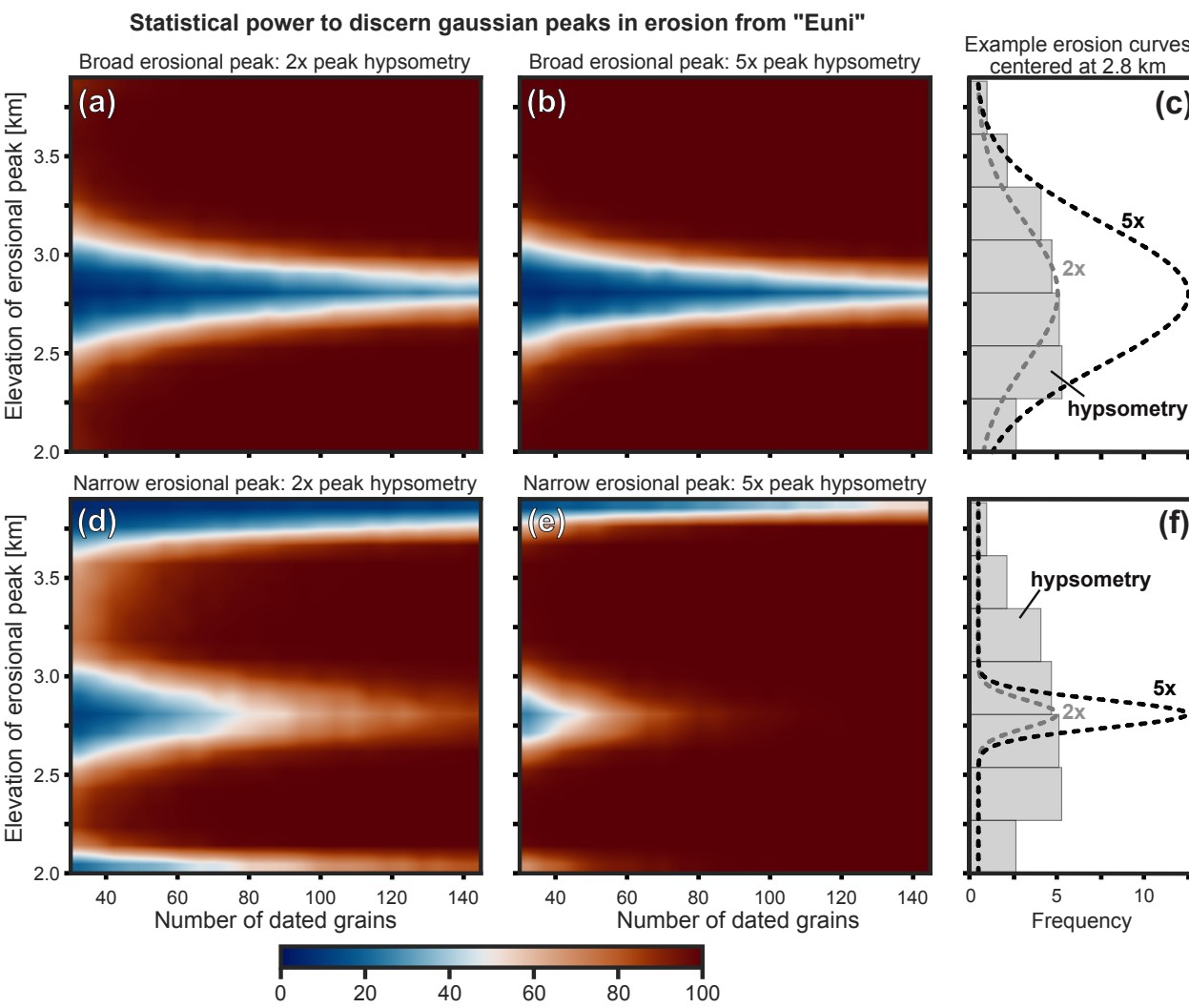

**Figure 10: Analysis of statistical power in discerning between the uniform erosion scenario 'Euni' and scenarios with a broad (a,b) or narrow (d,e) gaussian peak of erosion, as a function of sample size and elevation of the gaussian peak. The color scale informs the statistical power as a function of the number of grains (x axis) and the elevation at which the gaussian peak is centered (y axis). The catchment hypsometry (c,f) and age-elevation-relationship used here are the same as the Inyo Creek data (Stock et al., 2006). (a) Gaussian erosional function with $1\sigma = 500$ m and two times the peak prominence of the hypsometric histogram. (b) Gaussian erosional function with $1\sigma = 500$ m and five times the peak prominence of the hypsometric histogram. (d) Gaussian erosional function with $1\sigma = 100$ m and two times the peak prominence of the hypsometric histogram. (e) Gaussian erosional function with $1\sigma = 100$ m five times the peak prominence of the hypsometric histogram. (c,f) Example erosional functions for peaks centered at 2800 m elevation. The reader is referred to the text for more details.**

## 7 Other sources of uncertainty

The approach presented here for predicting and interpreting grain-age distributions enables quantifying the confidence level

in rejecting the uniform erosion hypothesis as well as the statistical power in discerning a prescribed erosion scenario from

uniform erosion, through detrital tracer thermochronology. Confidence level and statistical power are computed as a function of the sample size, the catchment properties and input erosion scenario. Nevertheless, even if the plausibility of a scenario is inferred to be high, our software, by design, does not suggest a unique solution and interpretations are also subject to additional sources of uncertainty. These include: (1) complex bedrock age-elevation relationship (Malusà and Fitzgerald, 2020); and (2)

spatial variability of sediment size resulting from either of transport distance (e.g. Lukens et al., 2020; Malusà and Garzanti, 2019), geomorphic process (e.g. Riebe et al., 2015; van Dongen et al., 2019), lithological differences (von Eynatten et al., 2012), or vegetation effects on weathering and erosion (Starke et al., 2020). Both factors should be considered as discussed here below.

In some cases, elevation alone cannot explain the entire variance of bedrock thermochronometric ages. For example, spatial variability of bedrock ages may reflect the proximity to tectonic structures or sub-catchment thermal events (e.g., magmatism, spatial changes in thermal gradients in proximity of major faults, etc). In such cases, an improved sampling of the spatial distribution of bedrock ages is needed, and may require more complex interpolation functions (e.g. Glotzbach et al., 2013). To accommodate these complexities, *ESD_thermotrace* allows for 1D and 3D-linear interpolation as well as radial basis function

interpolation. In all these cases, the interpolated age uncertainty is calculated through bootstrapping. In addition, users can opt to import an independently interpolated surface bedrock map such as from a thermokinematic model (e.g. Whipp and Ehlers, 2019). For additional information on bedrock thermochronometric age mapping the reader is referred to the "README" file included with the software (Madella et al., 2022). Regardless of which method is used, users should be aware that depending on the quality of the bedrock data, different interpolation methods may yield different predicted distributions. Accordingly,

resulting interpolated surfaces should be carefully evaluated, and a preference should be motivated by field observations and/or independent constraints. Here, the map of interpolated age uncertainty can also inform the locations where additional bedrock sampling would help reduce the uncertainty. Lastly, we note that an increase in age uncertainty always determines a decrease in the statistical power of the analysis.

Other possible sources of bias concern the grain size of the analyzed samples. Several issues may modify the original fingerprint of river sand (Malusà and Garzanti, 2019), such as downstream grain abrasion and fracturing, hydraulic sorting and weathering on the hillslope associated with a grain (Attal and Lavé, 2009). For example, grains sourced the farthest from the sampling spot may be underrepresented in the analyzed grain size fraction (Lukens et al., 2020), as has also been shown for the Inyo Creek catchment (Sklar et al., 2020). Furthermore, in addition to the mineral fertility inherent to the exposed bedrock,

the grain size distribution of the material found on the hillslopes (i.e. the material ready for transport) should be taken into account. Depending on the lithology (von Eynatten et al., 2012; Roda-Boluda et al., 2018) and on the locally dominant denudational process (van Dongen et al., 2019), different hillslopes of the same catchment may produce substantially different sediment size distributions (e.g. Riebe et al., 2015; Attal et al., 2015). Consequently, the mixed detrital sample may exhibit a bias in the relative abundance of the different age components. Both these issues can be mitigated through analysis of multiple

grain size fractions (Lukens et al., 2020), multiple measures of hillslope sediment size distributions, composite analyses of trunk stream and tributary stream sediment samples, and analyses of thermochronometers from different minerals.

## 8 Conclusion

This study reviewed previous approaches used to compare predicted and observed detrital grain-age distributions in the framework of tracer thermochronology. We have built upon these to develop a new tool (*ESD_thermotrace*) to investigate the
upstream pattern of catchment erosion and the confidence level in uniquely inferring this as a function of sample size and study-site-specific variables. To demonstrate the utility of this approach, we presented an analysis of previously published data from the Inyo Creek Catchment in California. The example highlighted the utility of measuring a large number of grains, and how multiple erosion scenarios are plausible for this catchment with the considered number of grains. The degree of statistical confidence permitted by this case study has also been quantified. We showed how the use of *ESD_thermotrace* can increase
the statistical rigor of tracer thermochronology studies and how it can also assist investigators in budgeting analytical costs of a future project. In cases where the number of datable grains is limited, the confidence level of the results can be quantified, and the statistical power of the analysis can be estimated.

**Code availability**

The source code of *ESD_thermotrace* (Madella et al., 2022) is freely available for download from the GFZ Data Services
(https://doi.org/10.5880/fidgeo.2021.003).

**Author contribution**

AM: conceptualization, formal analysis, investigation, methodology, software, visualization, writing - original draft preparation, writing - review & editing. CG: conceptualization, funding acquisition, project administration, writing - review & editing. TAE: funding acquisition, project administration, writing - review & editing.

**Competing interests**

The authors declare that they have no conflict of interest.

**Acknowledgements**

Research was funded within the DFG SPP Earthshape, through project numbers GL724/9-1 awarded to C.G., and EH329/18-1 awarded to T.A.E.. We thank M. Malusà as well as C. Lukens and C. Riebe for their constructive comments.

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

**Figure Captions**

**Figure 9**

Sketch of the difference between classical detrital geochronology (a) and tracer thermochronology (b). (a) Discrete age components are found in the detritus and refer to different upstream geological units. (b) A continuous detrital age distribution informs the relative abundance of material sourced from different elevations, based on a known age-elevation relationship.

**Figure 2**

Qualitative sketch to illustrate the effect of mineral fertility and erosion on the detrital distribution. (a) The catchment of Fig. 1b with known bedrock age (shades of green) is subject to 3 scenarios of spatially varying fertility and erosion. The box outlines refer to the curves below. (b) Detrital distributions obtained from the different scenarios in above. The green curve refers to spatially uniform erosion and fertility.

**Figure 3**

Example of metrics to compare two cumulative distributions drawn from different sample sizes. Here, an observed Cumulative Age Distribution (CAD) for $n=k$ (orange stepwise dashed line) is compared to a predicted CAD drawn for $n>>k$ (solid black line). The Kolmogorov-Smirnov statistic $d_{KS}$ equals the maximum absolute vertical distance between distributions in the observations-frequency space. The Kuiper statistic $d_{Kui}$ equals the sum of both positive and negative maxima. For $\alpha=0.05$ (i.e. 95% confidence) and $n=k$, a Dvoretzky-Kiefer-Wolfowitz confidence region can be calculated (gray shading). In this example, observed and predicted distributions are drawn from two statistically different populations at the 95% confidence level, because the orange curve exceeds the gray region.

**Figure 4**

Input bedrock data from Stock et al. (2006). (a) Raster images of the study area's digital elevation model, resampled to the user-specific cell size, and (b) bedrock surface AHe age interpolated based on the linear regression of age-elevation data. In both plots, point data inform bedrock sample locations and related AHe ages. Polygons show the location of the Inyo Creek catchment.

**Figure 5**

The raster image (a) displays the bedrock age interpolation error mapped to the topographic surface of the study area. The inset (b) shows the inverse variance-weighed linear regression employed as age-elevation function, as well as the associated prediction interval with $1\sigma$ confidence level.

**Figure 6**

Kernel Density Estimation curves of all predicted detrital distributions as well as of the observed detrital sample (a). Cumulative form (b) of the same distributions, plus the 95% confidence envelope (DKW-bounds) of 'Euni' for $n=k=52$. Euni: uniform erosion; E<Zmed: tenfold erosion below median elevation; E>Zmed: tenfold erosion above median elevation.

**Figure 7**

Confidence level at which the observed CAD (n=52) can be discerned from the uniform erosion prediction (cyan circle). The dashed cyan line shows how the allowed confidence level would vary, if the same observed CAD relied on 20-130 grain-ages. Statistical power of discerning the tested erosion scenarios from 'Euni' as function of sample size (solid lines). In this case study, the observed CAD of the Inyo Creek detrital sample allows rejecting the uniform erosion hypothesis with 55% confidence at best. An analysis to detect the scenarios 'E<Zmed' and 'E>Zmed' would have a statistical power greater than 95% in rejecting 'Euni' with more than 37 grain-ages. Euni: uniform erosion; E<Zmed: tenfold erosion below median elevation; E>Zmed: tenfold erosion above median elevation.

**Figure 8**

Violin plot showing the KS (and Kuiper) statistics between the predicted CAD of each erosion scenario and observed CAD, calculated for 1000 n=k=52 subsamples (a). KS (and Kuiper) statistics between the observed CAD and 1000 random n=k=52 distributions, drawn from the same detrital age population but including uncertainty (b). Every violin is split in two halves with equal area, each representing the probability distribution function of 10,000 KS (or Kuiper) statistics. The widest point of each semi-violin informs the median KS (or Kuiper) statistic; The closer to zero, the more similar is a scenario to the observed detrital age distribution. The most unlikely scenario is 'E>Zmed'. Both other scenarios are plausible, because their dissimilarity to the observed CAD largely overlaps the range of dissimilarity due to analytical error of the detrital ages.

**Figure 9**

Multi-Dimensional Scaling (MDS) plot. The 2-components MDS model fits a coordinate system to the measured dissimilarities (i.e. KS statistics) among all distributions. If the MDS model has a good fit, distances calculated in the new 2D-space (a) are a good approximation of the input dissimilarities among the elements. The goodness of fit is shown in (b), where modelled dissimilarities are plotted against input dissimilarities. For each distribution in (A) 68% and 95% confidence ellipses are also plotted.

**Figure 10**

Analysis of statistical power in discerning between the uniform erosion scenario 'Euni' and scenarios with a broad (a,b) or narrow (d,e) gaussian peak of erosion, as a function of sample size and elevation of the gaussian peak. The color scale informs the statistical power as a function of the number of grains (x axis) and the elevation at which the gaussian peak is centered (y axis). The catchment hypsometry (c,f) and age-elevation-relationship used here are the same as the Inyo Creek data (Stock et al., 2006). (a) Gaussian erosional function with $1\sigma = 500$ m and two times the peak prominence of the hypsometric histogram. (b) Gaussian erosional function with $1\sigma = 500$ m and five times the peak prominence of the hypsometric histogram. (d) Gaussian erosional function with $1\sigma = 100$ m and two times the peak prominence of the hypsometric histogram. (e) Gaussian

erosional function with 1σ = 100 m five times the peak prominence of the hypsometric histogram. (c,f) Example erosional functions for peaks centered at 2800 m elevation. The reader is referred to the text for more details.