# Peer review of "How many grains are needed for quantifying catchment erosion from tracer thermochronology?"

_Geochronology, 2021_

## Author Response (AR1)

**Response to reviewers' comments on *gchron-2021-6**

*All replies to comments are in italics.*

**Reviewer #1**

**General comment**

The authors present an open-source Python-based script for the analysis of detrital grain age distributions with respect to bedrock ages for tracer thermochronology. Although of potential interest for a readership dealing with thermochronology, the manuscript includes conceptual errors that should be amended before publication.

*We thank the reviewer very much for their time. We note that all the issues raised here deal with the importance of taking mineral fertility into consideration. We do appreciate these concerns and we will address them in the revised manuscript. Moreover, we are happy to see that these remarks do not highlight any conceptual error inherent to either of scope, method, results and conclusions of our paper.*

**Line-by-line comments**

lines 8-9: "If ages increase linearly with elevation, spatially uniform erosion is expected to yield a detrital age distribution that mirrors the catchment's hypsometric curve"

This statement is only true if mineral fertility is the same in any parts of the catchment, see Malusà, M. G., & Fitzgerald, P. G. (2020).The geologic interpretation of the detrital thermochronology record within a stratigraphic framework, with examples from the European Alps, Taiwan and the Himalayas. Earth-Science Reviews, 201, 103074.
Another important point is that ages are not always expected to increase linearly with elevation. Again, see Malusà and Fitzgerald 2020 - ESR about this point

*We thank the reviewer for this remark, we are aware of the importance of mineral fertility, which we have addressed in multiple occasions throughout the manuscript. We have use substituted the term "erosion" with "sediment production", however, for the sake of brevity and simplicity of the abstract we will not get into further details here.*

lines 33-37: "Geomorphologists have been able to infer changes in climatic parameters (Nibourel et al., 2015; Riebe et al., 2015), glacial erosional processes (Clinger et al., 2020; Ehlers et al., 2015; Enkelmann and Ehlers, 2015), sediment dynamics (Lang et al., 2018), relief evolution (McPhillips and Brandon, 2010), occurrence of mass wasting (Vermeesch, 2007; Whipp and

Ehlers, 2019) and differences in rock uplift (Glotzbach et al., 2013, 2018; McPhillips and Brandon, 2010)."

Unfortunately, in most of those cases mineral fertility was not taken into account, which makes the above conclusions invalid. I suggest integrating this part of the manuscript.

*We have added a remark along these lines to the revised manuscript. Please see lines 40-41.*

line 60: "If a range of assumptions hold (Malusà et al., 2013)"

Here, the authors may also quote Malusà and Fitzgerald 2020 – ESR where assumptions are discussed in detail

*We have added the suggested citation to the revised manuscript.*

Figure 2: What is the difference in mineral fertility between "low" and "high" in figure 2? In natural systems, fertility values vary within two or three orders of magnitude (see Malusa et al. 2016 Gondwana Research; Asti et al. 2018 Basin Res; Resentini et al. 2020 EPSL; Malusà and Fitzgerald 2020 – ESR), whereas these differences appear to be much lower in this figure.

*Figure 2 is a qualitative cartoon to illustrate how the combined effect of mineral fertility and erosion intensity affect detrital distributions. We have better specified this in the figure caption.*

lines 341-342: "Uncertainty in the interpretation can stem from factors such as: (1) complex bedrock age-elevation relationship"

This issue is addressed in detail by Malusà and Fitzgerald 2020 – ESR in their section 3, to which the reader should be referred to for the sake of clarity.

*We have added the suggested citation to the revised manuscript.*

lines 342-343: "(2) spatial variability of sediment size resulting from transport distance (e.g. Lukens et al., 2019)"

These aspects are addressed in much greater detail in Malusa and Garzanti 2019 – Springer, to which the reader should be referred to for the sake of clarity. Here is the full reference: Malusà, M. G., & Garzanti, E. (2019). The sedimentology of detrital thermochronology. In Fission-Track Thermochronology and its Application to Geology (pp. 123-143). Springer, Cham.

*We have added the suggested citation to the revised manuscript.*

lines 344-345: "lithological differences (von Eynatten et al., 2012), or vegetation effects on weathering and erosion (Starke et al., 2020)."

This is still part of the mineral fertility issue

*It is not clear to us what the reviewer is suggesting here.*

lines 360-361: "Other possible sources of bias concern the grain size of the analyzed samples. One issue is that downstream sediment abrasion may significantly modify detrital grain-age distributions, as can the weathering and erosion processes associated with a grain."

The problem is ill posed. The main issue is hydraulic sorting and selective entrainement rather than grain abrasion (see Malusa and Garzanti 2019 and Malusà and Fitzgerald 2020 and references therein for a discussion). Also, the impact of weathering is minimized by the single-mineral approach. I suggest rearranging the entire section.

*We have reworded this section for clarification and included the suggested citation. We think that the work of both mentioned authors (Lukens as well as Malusà) equally contribute to understand the grain size issue. We also think that weathering in the critical zone, especially for the case of studies relying on apatite U-Th/He geochronology is an important point to account for in this discussion. Please see lines 396-407.*

**Reviewer #2**

This is a joint review from Dr. Claire Lukens and Prof. Cliff Riebe.

*We are grateful for this very thorough and constructive review. The time the reviewers have taken to go through the manuscript and the code is much appreciated.*

**General Comments**

In their manuscript "How many grains are needed for quantifying catchment erosion from tracer thermochronology," Madella et al. seek to provide the tracer thermochronology community with a standardized open-access tool that 1) optimizes study design for efficiency in use of analytical funds and effort, and 2) quantifies limits on what can be interpreted from small sample sizes.

This work builds on previous applications of thermochronology in geomorphology. The open access software offers the ability to generate maps of variability in bedrock age, mineral fertility, and erosion rate from a specified DEM in a standard way, which makes it broadly useful. There is a commendable level of transparency and ease of implementation afforded by the availability of the model in a Jupyter notebook, which is sufficiently commented to make it widely applicable and adaptable. The manuscript also has easy-tointerpret and attractive conceptual figures that we could easily imagine appearing in a textbook chapter describing applications of tracer thermochronology.

*We are happy that the reviewers recognize the scope and potential of the paper.*

The overall approach uses bootstrapping to produce simulated (predicted) age distributions from simple erosion scenarios at the previously studied Inyo Creek catchment in California. It then compares each of these simulations to a measured dataset from the creek using a K-S or Kuiper test, and finally attempts to evaluate the resulting distributions of K-S (or Kuiper) test statistics to determine whether the predicted and observed distributions can be distinguished in a statistically meaningful way (we say "attempts to evaluate" because the hypothesis testing is flawed, as detailed below).

Using distributions of K-S test statistics this way - in a bootstrapping approach - is unconventional; though it's not incorrect, it is ultimately unnecessary in the manuscript as written, because it produces confidence intervals that can be calculated analytically using one of the equations they report in the text. However, it would be reasonable, valid, and more justified to use the bootstrapping method to extend these analyses to calculate the statistical power of observing effects of a given size with a given number of measured ages. The authors do not take that step here, but a comprehensive statistical power analysis would greatly strengthen a future version of the paper.

*We agree that, when testing scenario predictions against uniform erosion, Eq.3 can be used to calculate the Dcrit as function of sample size and confidence level. Based on which the minimum number of grains can be estimated. However, we find the bootstrapping approach necessary to treat additional uncertainties (e.g. analytical error, see detailed comments farther down below). An analysis of statistical power is a welcome suggestion, which we have implemented in the revised version of the manuscript.*

In addition to attempting to address the question "how many grains are needed" to distinguish different erosion scenarios, this manuscript presents some interesting examples in the discussion regarding the ability to detect differences from broad vs. localized patterns in higher erosion rates across the catchment. While these examples are illustrative and interesting, they are arbitrary, and thus do not provide a comprehensive analysis of what sorts of erosion patterns might actually be detectable. Moreover, the authors incorrectly argue that they have calculated the probability of detecting these patterns, which throws their interpretation into disarray.

*We also acknowledge we made poor choices of terminology to describe what our software can and cannot do. We will correct all instances in the revised version.*

The manuscript is broadly successful in arguing that tracer thermochronology studies should be designed with the size of the expected effect in mind. This will help guide the sampling strategy, including the number of ages that need to be analyzed. To the extent that the tool produced by the authors can streamline and standardize this in detrital thermochronology, it is commendable.

*We are happy to see the reviewers recognize the scope and the potential of this work.*

However, there are some major issues that make this manuscript unacceptable for publication as written. The most crucial of these are several statistical misconceptions and errors demonstrated by the authors, including an illegitimate chain of inference in the hypothesis testing. Perhaps most critically, there is a sense (lines 20-25 in abstract) that they expect the approach to be applied by future users to find best-fit erosion scenarios for their study sites; such results cannot be achieved using frequentist, statistical hypothesis tests of the kind employed here. While many of the points raised above might be readily fixed in revision, the fundamental errors in the statistical approach are quite serious and must be fixed for this work to move forward.

*In the revised version of the manuscript, we have better defined the scope and ability of our software. We agree that ESD_thermotrace is not suitable to find a best-fit scenario (yet), because it does not iterate through all possible ones. However, among other things, it allows quantifying how likely a detrital sample may have been drawn from the distributions predicted for each of the tested scenarios, based on random sampling of the KS test statistic. Please see e.g. lines 156-161*

**Major issues:**

**Problem Statement.**

The problem statement is poorly posed in the abstract and introduction, where it says: "However, there is no established method to quantify the sample-size-dependent uncertainty inherent to detrital tracer thermochronology, and practitioners are often left wondering 'how many grains is enough?'" But as the authors then later point out in equations 2 and 3, there actually is an established method, and it has been around for decades. Namely, Equation 3 can be readily applied to analytically solve for whether a difference between hypsometry and a measured detrital age distribution (as expressed by the K-S test statistic, D) would be significant at a specified alpha level for a given sample size. For example, given a false positive rate of 0.05 and a sample size of 100, D would have to be at least 0.136 for the analyst to be able to reject the null hypothesis that the age distribution arose from uniform erosion of the catchment. For a sample size of 50, D would need to be at least 0.192. The figure below shows how the least significant D (also known as Dcrit in some texts) varies with sample size.

[Figure]

Fig. R1. The least significant K-S test statistic, D, for a one sample hypothesis test decreases as one over the square root of sample size (number of grains measured). Results for alpha = 0.05 are shown. This shows how big the effect has to be able to reject the null hypothesis that the distributions are drawn from the same population as the hypsometric age distribution with 95% confidence or 5% false positive rate (where "effect" in a K-S test is the vertical offset between the measured CAD and erosion scenario).

Fig. R1 was created with very little effort using Equation 3, so it questions the whole premise of applying the much more complicated numerical approach that the authors use to tackle this problem. Note that a two-sample version of Equation 3 is also available, so a similar analysis could be employed to compare the hypothesis of uniform erosion against other specified erosion scenarios (or any two scenarios for that matter). Both the one sample and two sample problems can be found in undergraduate-level statistics textbooks. At minimum, the problem as posed is misleading, because its central premise is that no established methods are available. Perhaps more relevant to the value of the paper is that all the subsequent numerical effort seems unnecessary and overly complicated given that there actually is an established method on the books.

*We only partially agree on this point. Although Eq.3 has been well known for decades, we'd argue that it is not established in the detrital thermochronology community as means of*

*deciding the number of grains to be dated. As far as we can tell, also the reviewers, in their 2015 paper, have favored bootstrapping over the analytical solution of Eq.3 to treat uncertainties. The way we see it, is that Eq.3 does not account for additional uncertainties in the observed grain age distribution. What certainly is not established yet, is an open-source tool to apply both Eq.3 and bootstrapping to any case study.*

*The analytical solution yields a good approximation of the least significant dissimilarity as a function of sample size and confidence level (independently from the shape of the distribution). However, the probability of rejecting a scenario also depends on the distribution it is tested against (and related uncertainty). Therefore, Eq.3 may be used to solve one part of the problem (the least significant D for uniform erosion to be rejected), but random sampling of the other scenario (be it predicted or observed ages) is required to fully assess the confidence and the statistical power of an analysis. We have revised the manuscript to clarify this point and to better define the approach and novelty of our contribution. Please see e.g. section 5.4 of the revised manuscript*

Additionally, as one of the cited papers (Vermeesch et al., 2007) has pointed out, CADs in and of themselves (without the analytical uncertainties) are unbiased estimators of the population age distribution even when there are analytical errors on each measured age. This further emphasizes that there really is no need to employ bootstrapping to obtain the results presented here.

*To our understanding, the main reason why CADs (the sorted mean ages) are unbiased estimators of the population age distribution, is that the frequency of the measured age components is not affected by analytical uncertainties - which would increase the sampling of more precise ages. This does not imply that the probability of rejecting a scenario should not account for analytical uncertainties. In the revised version of the paper, to avoid introducing bias, we apply the mean relative uncertainty of the observed ages. We have updated the code accordingly.*

**Arbitrary (and therefore not generalizable) examples.**

The results of this work are not easy to generalize, because much of the paper focuses on evaluation of just three erosion scenarios: one with uniform erosion rates across the catchment; one with erosion rate doubling at 3000 m elevation; and one with erosion rate halving at 3000 m elevation. To a certain extent this is by design: the paper is definitely less about understanding erosion patterns and more of a vehicle for software that others could use as a tool to evaluate many additional cases (and thus strive for more generalized results). But therein lies perhaps the single biggest flaw in the paper: It encourages readers to use the code to go out, evaluate every scenario, and find the best fit. Unfortunately, that is not something that the frequentist statistical hypothesis testing methods employed here are capable of doing, as elaborated next.

*As the reviewers mentioned in this comment, this manuscript is not designed to discuss the erosional pattern of the Inyo Creek catchment, which is merely used as a testing ground for the proposed software. The arbitrary erosion scenarios were chosen for the sake of simplicity and for the sole purpose of illustrating how the software works. To improve this*

*aspect, we have rewritten the function that calculates the default test scenarios. Two erosional step functions of elevation are calculated, one with F-times higher erosion efficiency above Z and one with F-times higher erosion below the same Z. Here, Z equals the median elevation of the catchment. The factor F equals twice the prominence of the hypsometric peak. Please see related comment farther below and lines 238-245 for more details.*

*We acknowledge inaccurately stating that our code finds the "best-fit" scenario. The confusion here stems from the fact that we employ the KS test for "goodness-of-fit". We have revised the manuscript, stating that ESD_thermotrace finds the least dissimilar (i.e. most plausible) among the tested scenarios, using the KS test statistic as a metric for dissimilarity. Please see section 5.5.*

**Illegitimate chain of inference in statistical hypothesis testing.**

This problem starts on line 95, where the authors define the main aim of tracer thermochronology as finding "the best-fit pattern of erosion by minimizing the mismatch between observed and predicted distribution." Firstly, this is a very narrow assessment of the aim of tracer thermochronology. For example, the approach also has the potential to shed light on spatial variations in sediment size distributions, and many papers cited elsewhere in this text use tracer thermochronology to inform understanding of tectonics. The scope of this statement should be expanded. In addition, it needs to be restructured to be more in line with what the methods in the paper are able to accomplish. If the aim is as stated, "to find the best-fit pattern of erosion by minimizing the misfit between the observed and predicted distribution," then the frequentist statistical approaches outlined here are not appropriate.

*We have changed the contested statement, which, as the reviewers point out, is limited and not in line with the capabilities of our software.*

There is an excellent summary of what classical hypothesis testing can and cannot do at this site: https://www.envidat.ch/dataset/data-analysis-toolkits/resource/8c6c7021-0811-480f-ab52-00fbe3886591 (Toolkit 07). There are many things that this manuscript says it does that it cannot do based on these guidelines. The hypothesis testing in this manuscript is essentially set up backwards, in that it attempts to assess the "fit" of different scenarios to a measured distribution (something statistical hypothesis testing cannot do), rather than to reject the null hypothesis that the two distributions are drawn from the same distribution (which is what they are designed to do).

*We thank the reviewers for the excellent resource cited here. We acknowledge we used incorrect/confusing terminology. We have corrected the text not to falsely claim we assess the fit. Our software allows (i) quantifying the confidence level allowed by the sample size and the minimum number of grains to reject the uniform erosion hypothesis at a specified confidence level, (ii) given a number of grains, quantifying the statistical power of discerning erosion scenarios from uniform erosion, (iii) finding the erosion scenario that is most plausible (i.e. least dissimilar) to the observed distribution (among the tested scenarios). Please see section 5.4*

An example of the misconception of hypothesis testing occurs on line 147, with the phrase "rejecting or confirming a study's hypothesis". A statistical hypothesis cannot be confirmed; it can either be rejected with a specified false positive rate or it can be said that the data are consistent with the hypothesis. We call out many more examples of these illegitimate inferences in the numbered line-by-line comments below.

*We acknowledge the incorrect use of technical terminology. We have corrected these in the revised manuscript.*

To their credit, the authors are careful with interpreting their results, and do not claim that their findings require a reinterpretation of previous work in terms of erosion patterns or geomorphic processes. However, in their restraint, they also fall into another common pitfall with hypothesis testing -- the incorrect inference that an effect that cannot be rejected with 95% confidence is not meaningful at all. And perhaps the biggest concern is what happens when readers incorrectly use the code widely, as outlined in section 5.5, in an attempt to identify the "best fit" erosion rate relationship for catchments where detrital thermochronology can be applied.

*We regret giving the impression that inferences with confidence lower than 95% are meaningless. This is far from the message we aimed to convey. We are very much aware that in geology high confidence levels are often unlikely to be reached because of several reasons other than the study design. What we envision, is our contribution to be useful for all readers to quantify the confidence of their inferences easily and consistently.*

**Imprecise and/or incorrect language regarding statistics.**

The language throughout the manuscript is imprecise and in some cases incorrect or misleading regarding statistics. For example, in the abstract, the word "population" is used to refer to what, in the parlance of statistics, is a "sample." Tests that have a sufficiently large K-S statistic to reject the null hypothesis that the two samples came from the same population are incorrectly referred to by the authors as "successful," and the fraction of simulations where the null hypothesis is rejected are incorrectly termed by the authors a "success rate," especially in the documentation for the model code. In both cases, this is unconventional at best and will therefore probably also be misleading to many readers. While it is common to see phrases such as "we fail to reject the null hypothesis," in the literature, rejecting the null hypothesis is not, strictly speaking, a "success." It's just an indication that there's a sufficiently low likelihood of the observed effect arising by chance from conditions expressed in the null hypothesis. And crucially, the threshold for what counts as a sufficiently low likelihood is something the analyst decides on in advance (e.g., alpha = 0.05 in many studies)

*We have corrected the use of the term "success rate" and "population" accordingly.*

There is a particularly problematic issue on line 146. As written, it conflates the well known problem of post-hoc hypothesis testing, in which a set of measurements is compared against a hypothesis that was identified after the data were collected and analyzed (aka p-hacking), with what these studies actually did. To wit, they identified the null hypothesis —

of spatially uniform erosion rates — even before they went into the field. In addition to being an inaccurate statement of how these studies were conducted, the authors' text here has the possible effect of suggesting that the previous studies engaged in scientific misconduct, which we trust was not the intent.

*We regret giving the impression of suggesting that the cited previous studies engaged in scientific misconduct. It was not our intent. We maintain that due to the commonly low sample sizes and rare sample replication, geologic studies could often do better at quantifying uncertainties. With our work we propose a tool to do so in tracer thermochronology.*

This problem, and others like it, need to be fixed. We identified as many as we could find in the line-by-line comments below. These may seem like quibbles, but they illustrate the general pattern of confusing language that is inconsistent with established norms and will make it difficult for many readers to follow. It also undermines the credibility of the authors on the statistics, which is probably not a desired outcome in a manuscript that seeks to establish a widely used tool for statistical analyses of detrital age distributions.

*We acknowledge the incorrect use of technical terminology. What we hoped to convey as a more accessible language (avoiding statistical jargon on purpose) came across as misleading and incorrect terminology. We have corrected this issue throughout the text in the revised manuscript.*

**Linear Regression for age-elevation relationship**

The procedure for obtaining regression parameters and their uncertainties, by resampling the bedrock ages according to their errors and thus creating fits to 1000 different sets of bedrock ages, is non-standard and effectively presumes that all error in the regression is due to analytical uncertainties in ages. There is good evidence from the age elevation plot that this is not a valid assumption: Specifically, the error bars on the data points do not overlap the fitted line, indicating that, if there really is a linear relationship between age and elevation at the site, then there is variability in ages that is not accounted for by the analytical errors. Standard linear regression procedures in introductory texts (e.g. Helsel et al., 2020) outline more conventional approaches to quantifying uncertainty in regression parameters arising from scatter in x-y data, and free software is available to calculate these values. Slightly more complicated algorithms, also readily available (including in python), include procedures that invoke inverse variance-weighting in linear regression that could be used to account for differences in analytical error among measured ages. We recommend the authors use such a method in revision for more realistic estimates of regression uncertainty.

*Thank you for this useful suggestion, we have updated the code so that it performs inverse variance-weighting in the case of the linear regression interpolation method and related standard deviation. Please see section 5.1*

**Figures**

Figures 1 and 2 are both beautiful and helpful.

*Thank you.*

Figure 3. This is a useful figure. But since it is physically impossible to get a CAD to ever have a cumulative frequency <0 or >1, the vertical axes should only run between that range.

*We have limited the y axis to the 0-1 range in the revised version of this figure.*

Figure 4. There is a 10 m DEM for the region. Why not use that instead of the 30 m resolution?

*We use the 30-m-resolution because these figures are output from our software, therefore they employ the user-defined cell size. Running interpolations with a 10 m cell size would certainly yield better-looking models, but it would unnecessarily increase computational time.*

Figure 5. This is confusing. Why is percentage error shown here as opposed to absolute error? Also, see comment above about regression approach.

*We have changed this to absolute error, although we do not see why it should be confusing.*

Figure 6. Not sure why the thin purple lines do not match the expected step functions expected for individual simulated CADs based on Figure 3. Does this mean the authors are smoothing the CADs? If so, then, contrary to their goal, they have introduced the bias, despite their efforts, that Vermeesch (2007) warned about. As he points out, the CAD is an unbiased estimator of the true age distribution, without need for any smoothing. In fact, the smoothing is what introduces the bias, not the use of PDFs as the authors seem to suggest in line 237.

*Thank you for pointing this out, the thin lines are line plots that connect the mean ages. These are indeed treated as step functions in the code but appear smoothed in the plot. We have corrected the plots of the revised version for consistency.*

Figure 7 is unsatisfying, because it doesn't actually show the required number of measurements needed to detect the proposed erosion scenario. There should be a clear answer to the question posed in the title of the paper (How many grains…) here -- with a known number of measurements and a simple testable erosion hypothesis. Otherwise, readers are left wondering how many grains would be needed to get to 95% confidence. Why not run the same scenarios for more grains, until that threshold is reached? The erosion scenarios - doubling or halving erosion rates at 3000 m - are arbitrary and post-hoc anyway; why not choose one where we get an answer to the "how many grains" question? The authors do acknowledge on line 254 that these doubling and halving scenarios are not very different from uniform erosion, so the finding that 140 grains is not

sufficient should not be surprising. It leaves the reader wondering, how big of a departure from uniform erosion is needed to detect an effect with say 52 grains (as in Stock et al.) or 73 (in Riebe et al.)?

*We have changed the test scenarios to non-arbitrary ones (see related comment further up). The new erosion scenarios are step functions of elevation, with a F-times increase (decrease) in erosional weight at the median elevation of the catchment. The factor F equals twice the ratio between the most frequent and the least frequent elevation bin. For this calculation, the hypsometric histogram is constructed with a number of bins equal to the maximum difference in bedrock cooling ages, divided by their mean uncertainty, rounded up to the next integer.*
*In other words, these modified step-function scenarios, allow testing for two simple erosional patterns that are twice as pronounced as the hypsometric peak. Please see lines 238-245.*

Figure 8. The violin plots are not sufficiently explained in the text, and even after digging through the function code and commented scripts we are still not sure exactly how this plot is generated and what it means. Our take is this essentially yields a statistical power analysis. It is not recognized as such, and, moreover, the figure is incorrectly used to identify "fit," which this kind of hypothesis testing cannot do. Using a bootstrapping approach to determine the confidence on the K-S test statistics resulting from random sampling of the true age distribution is reasonable, but this spread in K-S could also be determined analytically using Eq. 3. A comparison of the analytical and bootstrapped findings in this case would support the overall approach.

*As already stated in this reply, we can't fully agree on this point. Although Eq.3 has been well known for decades, we'd argue that it is not established in the detrital thermochronology community as means of deciding the number of grains to be dated. As far as we can tell, also the reviewers, in their 2015 paper, have favored bootstrapping over the analytical solution of Eq.3 to treat uncertainties. The way we see it, is that Eq.3 does not account for additional uncertainties in the observed grain age distribution. The analytical solution yields a good approximation of the least significant dissimilarity as a function of sample size and confidence level (independently from the shape of the distribution). However, the confidence of rejecting a scenario also depends on the distribution it is tested against (and related uncertainty). Therefore, Eq.3 may be used to solve one part of the problem (e.g. the least significant D for a null hypothesis to be rejected), but random sampling of the other CAD (be it predicted or observed) is required to fully assess the confidence of the analysis. We have revised the manuscript to clarify this point. Furthermore (see also reply to comment referring to line 271), the percentages in Fig.8 inform the false negative rate: how often one would be unable to reject the $H_0$ that two samples are taken from the same distribution (the respective erosion scenario), even if they actually weren't.*
*While we agree that it is incorrect to label these as "degree of fit", they do characterize the plausibility of each tested scenario, so we'd rather maintain this term for simplicity.*

Figure 9 is useful, and provides a nice visual representation of the comparison between the observed Inyo Creek data and the two proposed erosion scenarios. However, the MDS

approach isn't well explained; including a bit more information for those unfamiliar with the approach would be useful. For example, what goes into each of the MDS parameters on the axes?

*The cited work by Vermeesch (2013) is an exhaustive resource to understand how this model works, we have better introduced the principles of MDS in the revised version but will not expand on the details of this method.*

Figures 10 and 11 both incorrectly state that the color shade refers to the "probability to discern scenarios from 'Euni.'" The probability of detecting an effect of a given size when it is actually present (and thus correctly rejecting the null hypothesis) is known as the statistical power (aka true positive rate), equal to one minus the false negative rate. However, what the authors have been calculating thus far, e.g., in Figure 7, is the confidence level on K-S values, which is a statement about the least detectable K-S statistic, D. The minimum detectable difference, which yields insight about statistical power, is not something the authors have characterized here. So these are not probabilities, but confidence levels. The distinction is really important. In a t test, for example, which is in a lot of ways similar to the K-S test (at least in how it is applied), if the true mean is right at the threshold of detection with 95% confidence (i.e., with a 5% false positive rate), the chance you will correctly reject the null hypothesis is only 50%, because the mean is right at the threshold and half of the t distribution is on the rejection side of the threshold. This seeming paradox is actually just a demonstration of the fact that power and confidence are two very different things. Hence, the authors cannot call the color bar the probability of detection.

*We have merged Figures 10 and 11 and corrected the terminology in the revised version. The color shades in Figure 10 inform the probability that a Gaussian peak of erosion produces a detrital distribution whose dissimilarity (the KS statistic compared to Euni) is greater than the least significant dissimilarity allowed by the sample size.*
*To our understanding, this does represent the statistical power of detecting a hypothetical Gaussian peak of erosion whose effect size is pre-assigned.*
*Please see section 6 and Fig.10.*

**Line item and specific comments (note: some of these are quick fixes but some are very substantial):**

Line 8. If bedrock ages increase…

*We have corrected this in the revised version.*

L9. "Mirror" suggests mirror symmetry, which is not what you mean. Use "matches" instead? Or "closely follows?"

*We have corrected this in the revised version.*

L10. Another thing this may indicate is that sediment size distributions vary across the catchment and the collected sample is not representative. This issue, which is at least as important as the mineral fertility issue which is included explicitly in the code (and demonstrably more important in Inyo Creek), is not addressed until the discussion and then in a way that does not clearly state the potential for bias.

*We have better specified all sources of potential signal modification throughout the text, while here in the abstract, we have substituted "erosion" with the more general "sediment production". Please see e.g. lines 10-11*

L11. In the parlance of statistics, "population" refers to the group from which "samples" can be drawn. So a set of measured ages from grains collected from a stream is a sample of the population, not a population.

*Thank you, we are aware of this. The use of the term "population" was intended to distinguish between the material collected in the field (sample) and the statistical sample. However, we have corrected this throughout the text to improve the statistical rigor of this work.*

L11. Also, "measured grain-age populations" is a noun train that is hard to understand. What is a "grain-age population?" Also, strictly speaking it's not the age of a grain, it's the cooling age of the grain.

*We have changed "measured grain-age populations" to "detrital samples" . Considering that we have introduced the thermochronologic context of this paper, we find it reasonable to use age instead of thermochronometric age, for simplicity.*

L12-13. Yes, discerning differences can be difficult. But this statement misses an important qualifier about how different the scenarios are. If the two erosion rate patterns under consideration are not very different, then small sample sizes can be a problem. If, on the other hand, the differences are substantial, then discerning between two different patterns may not be problematic at all, even for small sample sizes. Equation 3 in the text shows this: The size of the "least significant" K-S statistic D (also known as Dcrit in some texts) scales as sqrt(ln(2/alpha)/(2*k)) or 1.36/sqrt(k) for alpha = 0.05 where k, in statistical parlance, is the "least significant number." So, if the effect is big, even a small sample size can detect it.

*We have added the effect size to this statement in line 12*

L13-15. This is not true. In fact, Equation 3, presented later on line 131, which is so well established one can read about it in Wikipedia, can be readily applied to analytically solve for whether a measured difference in two detrital age distributions would be significant at a specified alpha level for a given sample size.

*As already stated above, we partly disagree on this point. Although Eq.3 has been well known for decades, we'd argue that it is not established in the detrital thermochronology*

*community as means of deciding the number of grains to be dated. The way we see it, is that Eq.3 does not account for additional uncertainties in the observed grain age distribution. What certainly is not established yet, is an open-source tool to apply both Eq.3 and bootstrapping to any case study.*

*The analytical solution yields a good approximation of the least significant dissimilarity as a function of sample size and confidence level (independently from the shape of the distribution). However, the confidence of rejecting a scenario also depends on the distribution it is tested against (and related uncertainty). Therefore, Eq.3 may be used to solve one part of the problem (the least significant D for uniform erosion to be rejected), but random sampling of the other scenario (be it predicted or actual observed ages) is required to assess the confidence of the analysis. We have revised the manuscript to clarify this point and to better define the approach and novelty of our contribution. Please see section 5.4.*

L15. What is meant here by "enough?" Enough to do what? To detect an effect of a given size?

*We have better clarified this point in lines 15-16.*

L94. In addition to this list, the authors need to add something about variability in sizes of sediment produced on slopes and whether the size class sampled in the stream is representative of erosion from the catchment. There are now multiple papers that show this is at least as important as the mineral fertility issue (listed here as item III). E.g., see Vermeesch 2007; Riebe et al., 2015; Lukens et al., 2020.

*We have addressed this point in the mentioned list too and referred to the suggested literature. Please see lines 102-104.*

L95. This is a very narrow assessment of the aim of tracer thermochronology. As we have proposed in several papers now, it has the potential to shed light on not just spatial variations in erosion rates but also spatial variations in sediment size distributions. In addition, as laid out in the Ruhl and Hodges and Vermeesch papers cited elsewhere in the text, tracer thermochronology has been used to inform understanding of tectonics as well. So the scope of this needs to be greatly expanded. In addition, it needs to be restructured to be more in line with what the methods in the paper are able to accomplish. If the aim is as stated, "to find the best-fit pattern of erosion by minimizing the misfit between the observed and predicted distribution," then the frequentist statistical approaches outlined here are not appropriate.

*We have expanded the statement and rephrased to avoid suggesting our software is able to find a best-fit scenario. Please see lines 107-109.*

L136. While "quantiles" is strictly ok, "percentiles" would be the more conventional and easier to understand term to describe the 2.5th and 97.5th cut points of the distribution.

*Corrected*

L144. This is incorrect. The past studies represent tests of a specific null hypothesis — i.e., that erosion rates are spatially uniform. That does not make them semi-quantitative much less qualitative. They are quantitative in their evaluation of a specific null hypothesis.

*We agree that rejecting $H_0$ (uniform erosion scenario) as population of the detrital sample is a quantified result. However, we maintain that testing multiple erosion scenarios would yield a further quantification of how likely specific erosion patterns can be discerned from it.*

L145. There is nothing wrong with assessing results with respect to a null hypothesis as long as the null hypothesis is established in advance. It does not "undermine the statistical rigor" of the studies, as the next sentence falsely states.

*We have corrected this statement. There certainly is nothing wrong with it. Please see lines 150-152.*

L146. Correct this, or strike it. As written, it conflates the well known problem of post-hoc hypothesis testing, in which a set of measurements is compared against a hypothesis that was identified after the data were collected and analyzed, with what these studies actually did. To wit, they identified the null hypothesis — of spatially uniform erosion rates — even before they went into the field.

*We have corrected this in the revised version of the manuscript. Please see lines 150-154.*

L147. "...rejecting or confirming a study's hypothesis..."  This is one example of how the manuscript currently turns legitimate inference on its head. A hypothesis cannot be confirmed using the kind of frequentist statistical hypothesis tests employed here. The observed data can be consistent with a particular hypothesis, but that doesn't make it "true." To get a handle on the probability of whether a hypothesis is true or not, it is necessary to take a Bayesian approach, which is well outside the scope here.

*We agree with this comment, what we meant to say is "reject or accept at the significance level…". We have corrected the inappropriate wording. Please see line 154.*

L180. This doesn't make sense. Again, there is a problem here with the chain of inference. It is not that you are trying to "detect" different erosion scenarios. You can't do this with frequentist approaches like a K-S test. You can identify differences that are unlikely to arise by chance from a proposed null hypothesis. But you cannot detect a specific scenario by finding a best fit, contrary to what the authors repeatedly state throughout the paper.

*We agree with the reviewers here. To us, "detecting" a scenario is the short version of what they suggest in their comment: "identify a dissimilarity that is unlikely to arise by chance". Our software is designed to quantify both of them (dissimilarity and likelihood). We have avoided using implicit and confusing terminology in the revised manuscript. Please see line 154-161.*

L187. It is not clear how 6 is different from 5. Try rephrasing either or both?

*We have rephrased these items to better clarify. Please see lines 189-208.*

L204. Inferred (past tense)

*Corrected*

L211. Usually reserve upper case $R^2$ for coefficient of multiple determination (in multiple regression) and lower case $r^2$ for coefficient of determination (in simple linear regression, as is the case here).

*Corrected*

L224. The attribution to Riebe et al. is incorrect. The bulk geochemical work that supports this observation was done by and reported in Hirt (2007).

*Corrected*

L249. "the inherent noise (i.e. dissimilarity)" is nonstandard terminology. In what sense is it actually "noise." And how does that then equate to "dissimilarity?" We suggest sticking to more traditional terminology to aid understanding for the broadest possible audience. In this case, the thing being referred to is the 95% confidence interval on the K-S statistic D. It's not the inherent noise.

*We have corrected the inappropriate terminology*

L264. This entire section is deeply flawed in that it suggests that users of the code can employ it to find a "best fit" erosion model to their data. As repeated multiple times in this review, that's not what statistical hypothesis testing can do.

*As stated above, we have modified the text according to the reviewers' comments on this issue.*

L269. As pointed out by Vermeesch (2007) this kind of analysis results in "double smoothing" of CADs (his term). He shows that this introduces the very bias that the authors say they are trying to avoid according to them in line 237.

We *thank the reviewers for pointing out this possible issue. In order to account for analytical uncertainty, we iteratively calculate the dissimilarity based on n=52 CADs obtained through random sampling of ages from their analytically measured distribution. To avoid the bias arising from applying variable 1sigma to the ages, in the revised version of the manuscript we apply the same average 1sigma to all of them.*

L271. The next two sentences and associated analyses are especially flawed. This so-called "plausibility" actually has a technical name in statistical hypothesis testing. It's the

false negative rate (beta), which describes how often you would fail to reject the null hypothesis that the two distributions are drawn from the same population when they actually are not from the same population. The "reliability" or "statistical power" of the test in detecting effects of a given size is equal to one minus beta. It's the complement to the false positive rate. So this reporting of 62.9% and 87.6% and 3.3% as a degree of "fit," rather than as the false negative rate, is completely at odds with the terminology, usage, and proper chain of inference in frequentist statistical hypothesis testing.

*We have corrected the inappropriate terminology here. The percentages in Fig.8 inform the likelihood that a random n=52 sample of each erosion scenario is as or less dissimilar to the observed CAD than the 95% confidence dissimilarity arising from the CAD's analytical uncertainty. Therefore, as the reviewers correctly understood, these percentages inform a false negative rate: how often one would be unable to reject the $H_0$ that the two samples are taken from the same distribution (the respective erosion scenario), when they actually are not.*
*Yes, it is incorrect to label these as "degree of fit", but in fact they characterize the plausibility of each tested scenario. It remains clear, however, that several scenarios may be equally plausible and the data may not suffice to further discriminate. We have better explained what these numbers and the violin plot show, but we will maintain the term "plausibility" because it is a more accessible term, as opposed to "false negative rate, beta". Please see section 5.5 and Fig.8.*

L284. Again, identifying a satisfactory fit is not something this kind of statistical hypothesis testing can do. A Bayesian approach could do it, but not a frequentist one like this application of K-S tests.

*We agree on this point, see comment just above.*

L292. "uniquely detect any of the tested scenarios at with confidence (Fig.7)" Typo aside (strike the "at"), this is not something that frequentist hypothesis testing can do. To appreciate this, it may help to realize there are only two tested scenarios, while there are infinite possible scenarios. "Uniquely detecting" a scenario would require testing all of the possible scenarios (which is of course impossible), not just two that have been arbitrarily chosen. The choice of an alpha of 0.05 is arbitrary, too - it's what is often settled on as an acceptable false positive rate (and even this is a subject of heated debate in the stats community). In any case, when you are unable to reject the null hypothesis at that 0.05 (i.e. threshold) false-positive rate it does not mean that you have 95% confidence that the scenario matches the observations. That would be akin to turning the logical inferences that can be made from these kinds of tests on their heads.

*We agree on this point too, we have corrected this.*

L297. As we suggested in Riebe et al., 2015, sediment size changes across this catchment. In addition, in Sklar et al., 2020 we also documented downvalley fining in sediment size on slopes in Inyo Creek. That work should definitely be cited here. Changes in sediment size are certainly beyond the scope of the manuscript presented here, but may provide an alternate explanation for deviations from non-uniform sediment production (not

just erosion, but also variations in sediment size). In fact, this is precisely what Riebe et al suggested: it was the greater-than-expected contribution from low elevations and lesser-than-expected contribution from higher elevations shown in Stock et al's fine sediment -- coupled with the opposite pattern in the coarse gravel -- that showed this. So the same pattern that the authors point to here (where the evidence from Stock et al.'s sample seems to point to greater contributions from the lower part of the catchment) is part of the basis for the conclusions of Riebe et al.

*Thank you for this comment, we have better discussed the potential grain size bias in the revised manuscript. Please see line 324-330.*

L319. If the authors are explicitly including age uncertainties in their analysis without adjusting for the resulting double smoothing described by Vermeesch (2007), then they are introducing the bias he discusses and that they earlier said they would avoid.

*As stated further up, we thank the reviewers for pointing out this possible issue. In order to account for the uncertainty of predicted age distributions, we follow the procedure to construct a "$CSPDF_{t'm}$" (same as CAD) as described by Vermeesch (2007) in his Table 1.*

L344 (and elsewhere): the Lukens et al. paper is officially 2020, not 2019.

*We have corrected this*

L365: More suggested citations for slopes producing different size distributions: Sklar et al. 2020 (ESPL) (at Inyo Creek); 2017 (Geomorphology) (generally); Roda-Boluda et al. (2018, ESPL) (related to lithology); Attal and Lave (2015) (related to erosion rate); Marshall and Sklar (2012) (related to climate)

*Thanks for the suggestions, we have included most of them in the revised manuscript.*

---

## Author Response (AR2)

**Response to the associate editor's line-specific comments on gchron-2021-6**

*All replies to comments are in italics.*

**Technical corrections:**

1. Line 14: "no established tool" to "no established software tools" to highlight this as a software contribution and distinguish it from the algorithms (also tools) used by the other practitioners and studies you cite.
*Done*

2. Line 112: change 'successively' to 'later'.
*Done*

3. Figure 4 caption: "Polygons show…"
*Done*

4. In Figure 5, I know of no (weighted) linear regression algorithms that yields a bumpy uncertainty envelope. Please verify that the spurs in the gray region of the inset are not an artifact of plotting/export code.
*We did thoroughly check this. The uncertainty envelope of a linear regression can inform either of (1) the standard error of the mean of predictions (smooth); (2) the confidence interval for the mean of predictions, at the chosen significance level alpha (smooth, probably the most common); (3) the prediction interval for future observations, at the chosen significance level alpha (bumpy). We employ the latter, more conservative and bumpy uncertainty envelope, at 1 sigma confidence. We have clarified this in the caption of Figure 5.*

5. Define statistical power before using it in section 5.4 and Figure 7.
*Done, please see lines 264-268 of the corrected manuscript*

6. Start a new paragraph at line 273, at "If the confidence…" to distinguish the discussion of power from the discussion of 'confidence'.
*Done*

7. Throughout manuscript: when grain ages and uncertainties are needed as input or used in a calculation, additionally include the word 'uncertainties' for clarity.
*Done*

8. Line 284: I believe this is correct, but please be more clear with the 'detect such scenarios' phrase by defining 'detect' and specifying which scenarios (i.e., the tested E> or < Zmed scenarios). This will help readers avoid easy misconceptions.
*Done, please see lines 289-292 of the corrected manuscript.*

9. Line 300: I think the term 'plausibility' is used appropriately (and colloquially) in lines 293 and the Figure 8 caption, but as Lukens and Riebe suggest, the term "false negative rate" should be included alongside the explanation here of how it's calculated.
*Done, please see lines 310-311 of the corrected manuscript.*

10. Line 343: Spell out the words in "…equals two and five times…", and likewise "five times" in line 344.
*Done*

11. Line 346: insert a comma after functions.
*Done*

12. Figure 10 caption: replace the numbers 2 and 5 with "two" and "five".
*Done*

13. Line 372: "statistical" instead of "statical"
*Done*

14. Line 373: "sample size" instead of "samples size"
*Done*

15. Line 417: delete "by any reason".
*Done*

16. Line 421: Note that the first review was signed anonymously.
*Yes, but he came out at a conference shortly after the review got to me, so I thought I could include his name too.*